# Optimizing potassium polysulfides for high performance potassium-sulfur batteries

Wanqing Song[1], Xinyi Yang[1], Tao Zhang[1], Zechuan Huang[1], Haozhi Wang [1,2] ✉, Jie Sun[3], Yunhua Xu[1], Jia Ding [1] ✉ & Wenbin Hu [1,4] ✉

Potassium-sulfur batteries attract tremendous attention as high-energy and low-cost energy storage system, but achieving high utilization and long-term cycling of sulfur remains challenging. Here we show a strategy of optimizing potassium polysulfides for building high-performance potassium-sulfur batteries. We design the composite of tungsten single atom and tungsten carbide possessing potassium polysulfide migration/conversion bi-functionality by theoretical screening. We create two ligand environments for tungsten in the metal-organic framework, which respectively transmute into tungsten single atom and tungsten carbide nanocrystals during pyrolysis. Tungsten carbide provide catalytic sites for potassium polysulfides conversion, while tungsten single atoms facilitate sulfides migration thereby significantly alleviating the insulating sulfides accumulation and the associated catalytic poisoning. Resultantly, highly efficient potassium-sulfur electrochemistry is achieved under high-rate and long-cycling conditions. The batteries deliver 89.8% sulfur utilization (1504 mAh g$^{-1}$), superior rate capability (1059 mAh g$^{-1}$ at 1675 mA g$^{-1}$) and long lifespan of 200 cycles at 25 °C. These advances enlighten direction for future KSBs development.

The development of efficient energy storage systems (ESS) is one key initiative for pursuing carbon neutrality worldwide[1]. Based on the high capacity of sulfur redox[2,3] and the low potential of potassium metal[4], potassium sulfur batteries (KSBs) have desirable theoretical energy density of 1023 Wh kg$^{-1}$ thereby holding great promise for next-generation ESS. KSBs are intrinsically low in cost due to the abundance of sulfur and potassium. Nonetheless, some challenges around sulfur cathode significantly plague the KSB operation, such as the low sulfur utilization, the sluggish kinetics of potassium polysulfides (KPSs) conversion, the severe shuttling, and the difficult decomposition of solid end-products (K$_2$S$_x$, $1 \le x \le 3$)[5]. These intractable impediments limit the capacity and lifespan of KSBs. Therefore, it is of vital importance to develop functional hosts accommodating the sulfur for addressing these issues.

Some pioneering efforts innovated functional sulfur hosts for KSBs. Most of the hosts are carbon-based, taking advantage of the good electrical conductivity, diverse surface chemistry, and versatile porosity of carbon[6,7]. The sulfur confinement is the primary function of the host. Physically confining sulfur in the carbon porosity is a popular strategy[8,9]. For instance, microporous carbons confining small molecular (S$_{1-3}$) exhibited promising capacity and cyclability in KSBs. Nonetheless, the limitation of pore volume normally sacrificed the sulfur loading[7,10]. The carbon hosts employed were normally decorated by nitrogen moieties, which are effective of enhancing the sulfur anchoring via covalent bonding at heterointerfaces[11]. In addition, the chemical confinement strategy was also widely reported[12–14]. Sulfurized polyacrylonitrile (SPAN) utilizing the covalent sulfur-carbon bonds greatly inhibited KPSs shuttling[14–17]. The chemical confinement also

[1]School of Materials Science and Engineering, Tianjin Key Laboratory of Composite and Functional Materials Key Laboratory of Advanced Ceramics and Machining Technology (Ministry of Education), Tianjin University, Tianjin, China. [2]School of Materials Science and Engineering, Hainan University, Haikou, China. [3]School of Chemical Engineering and Technology, Tianjin University, Tianjin, China. [4]Joint School of National University of Singapore and Tianjin University, International Campus of Tianjin University, Binhai New City, Fuzhou, China. ✉e-mail: hzwang001@tju.edu.cn; jiading@tju.edu.cn; wbhu@tju.edu.cn

includes the chemical adsorption between KPSs and certain polar species of the host, such as heteroatom groups, nanostructured compounds, atomic metal assemblies[7,18,19]. For instance, single metal atoms[18,20,21] and metal carbides[22–25] were reported providing adsorption effect towards polysulfides.

The function of catalytic conversion towards KPSs in sulfur host attracted increasing attention in recent years. Some highly enlightening studies introduced catalytic sites in the hosts to improve the KPSs conversion kinetics, especially the solid-state reactions including $K_2S$ oxidation[26]. Metallic single atoms[18,26] and metallic atom clusters[27] were reported as electrocatalytic species, which suppressed the long-chain KPSs shuttling and reduced the polarization of solid-state sulfide conversion, thereby enhancing the sulfur utilization, rate and cycling capability. The design of catalytic materials for KPSs can be inspired by the prior achievements in Li/Na-sulfur batteries. The metal element in various forms (e.g. single atom, atomic cluster, compounds) that can deliver general catalytic capability towards both lithium and sodium polysulfides deserves particular attention. For example, tungsten is catalytically active in Li-S batteries (tungsten single atom[20], tungsten carbide[23–25], tungsten sulfide[28,29]) and Na-S batteries (tungsten

nanoparticle[30]). The successful applications of tungsten-based catalysts in metal sulfur batteries were largely ascribed to the special electronic structure of tungsten atomic assemblies and tungsten compounds[31]. These insights are highly instructive for KSBs explorations.

The previous researches reveal the importance of introducing catalytically active sites in sulfur hosts for KPSs conversion. Considering the complex potassium-sulfur electrochemistry, more factors should be considered beyond the catalytic conversion. First, sulfur is essentially confined in the carbon porosity, which distributes ubiquitously in the host. On the contrary, the isolate nanostructured catalytic species can only locate disjunctively[19,32]. Therefore, there is an inherent mismatch in the distribution of sulfur and catalytic sites, which makes the KPSs migration among the catalytic sites highly necessary. More importantly, the reduction products of solid-state KPSs conversion are highly insulating and difficult to decompose[33], which can easily accumulate over the catalytic sites and cause catalytic poisoning[34,35]. Therefore, the migration of $K_2S/K_2S_2$ away from the catalytic sites is important for preventing the expansion of electronically inactive areas in the cathode, especially under high current density and aggressively

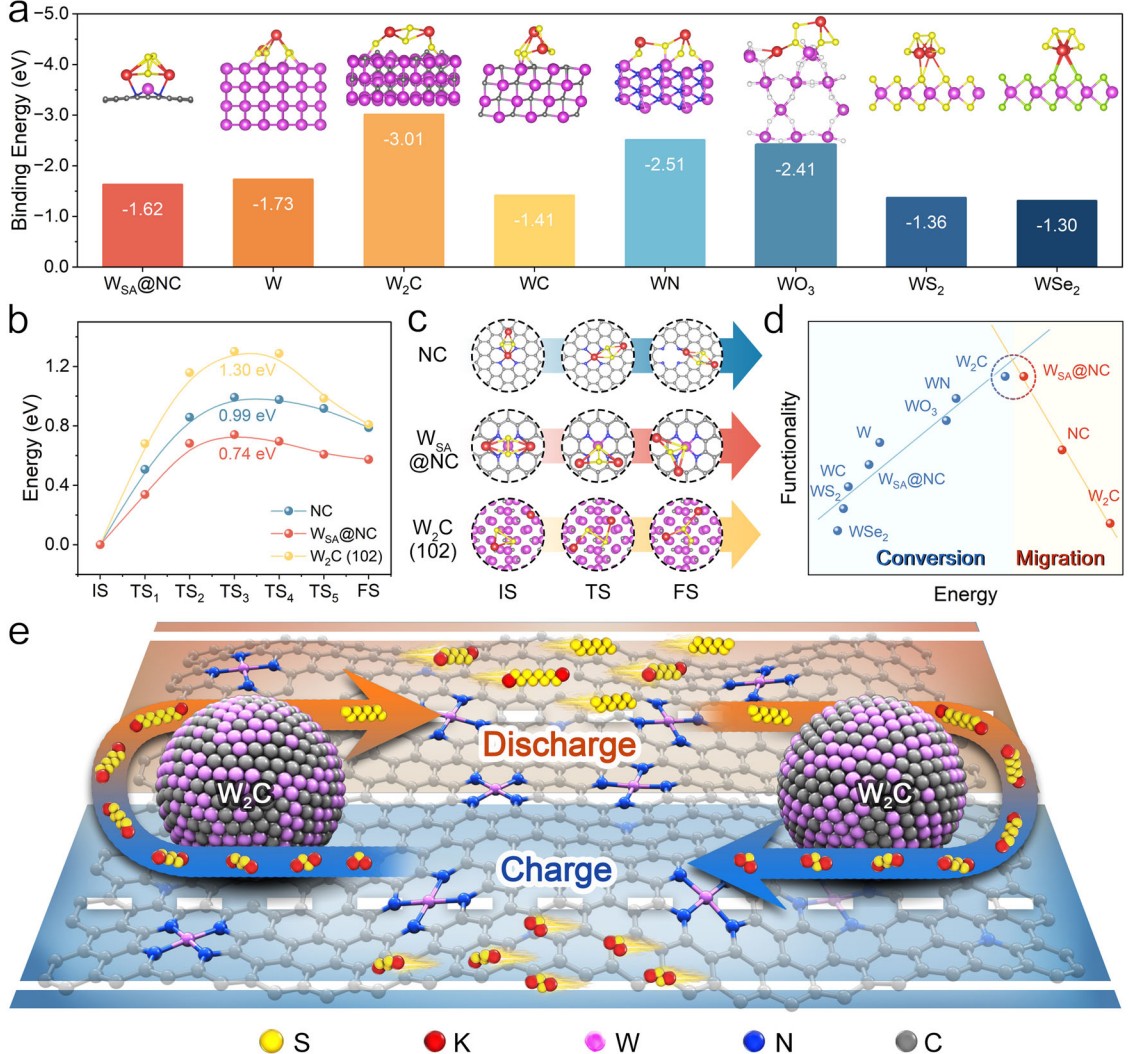

**Fig. 1 | Theoretical guidance and screening for sulfur host design. a** The binding energies of $K_2S_4$ on $W_{SA}@NC$, W metal (110), $W_2C$ (102), WC (101), WN (112), $WO_3$ (100), $WS_2$ (002) and $WSe_2$ (006) and the corresponding configurations (insets). **b** The energy profiles of $K_2S_2$ migration on NC, $W_{SA}@NC$, and $W_2C$ (102). **c** The calculated initial, transition and final states of $K_2S_2$ migration. **d** Volcano plot of KPS conversion and migration functionalities with respect to specific energy values. **e** Schematic demonstrating the design principle of the $W_{SA}$-$W_2C$ composite based sulfur host for efficient sulfur redox in KSBs guided by the theoretical calculation. Source data are provided as a Source Data file.

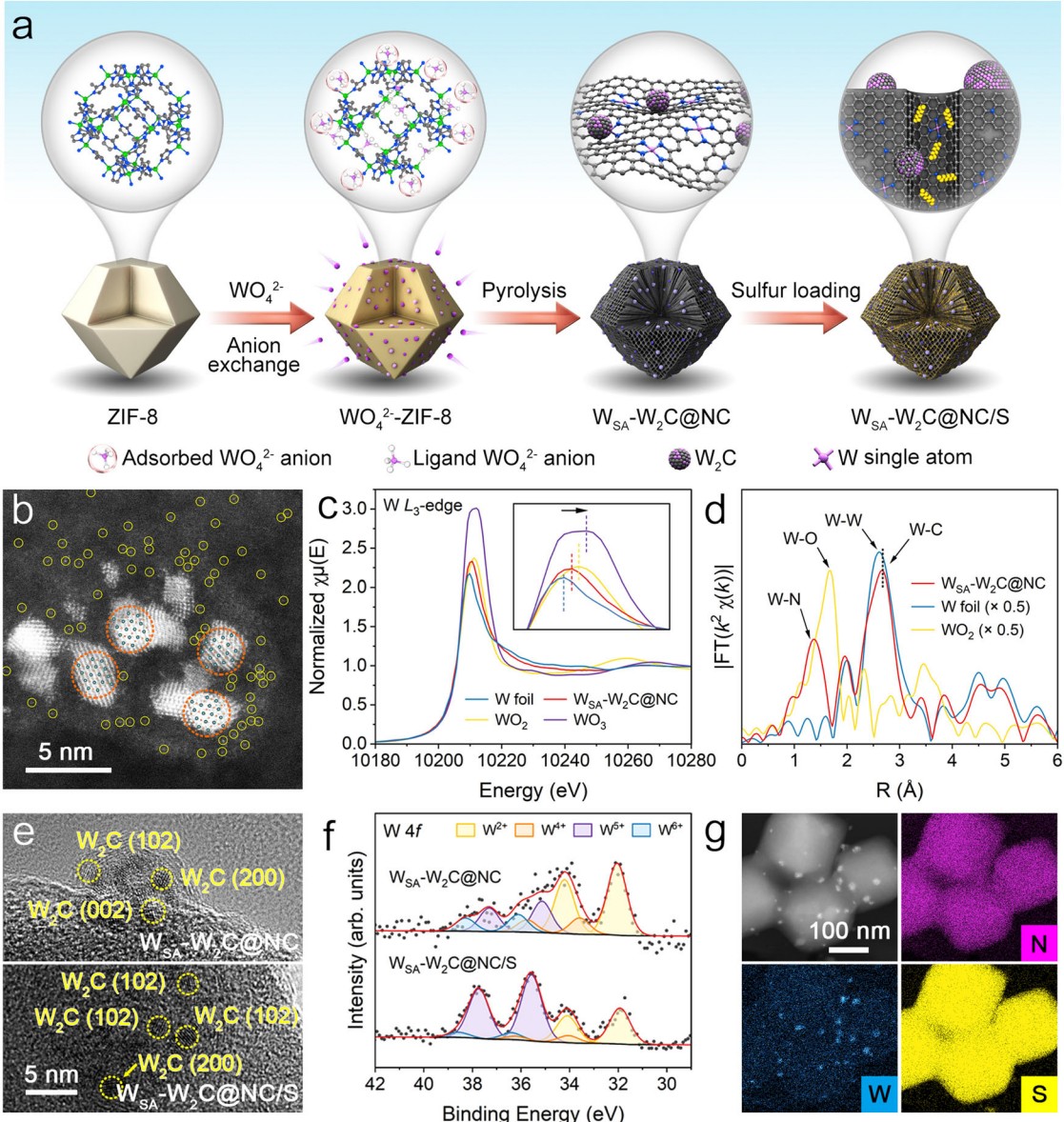

**Fig. 2 | Synthesis and characterization of $W_{SA}$-$W_2C$@NC and $W_{SA}$-$W_2C$@NC/S.**
**a** Schematic synthesis process of $W_{SA}$-$W_2C$@NC and $W_{SA}$-$W_2C$@NC/S. **b** HAADF-STEM image of $W_{SA}$-$W_2C$@NC. **c**−**d** XANES (**c**) and EXAFS (**d**) spectra of W $L_3$-edge for $W_{SA}$-$W_2C$@NC and W foil, $WO_2$, $WO_3$ references. **e** HRTEM images of $W_{SA}$-$W_2C$@NC and $W_{SA}$-$W_2C$@NC/S. **f** High-resolution W $4f$ XPS spectra for $W_{SA}$-$W_2C$@NC and $W_{SA}$-$W_2C$@NC/S. **g** HAADF-STEM image and corresponding EDS mapping of $W_{SA}$-$W_2C$@NC/S. Source data are provided as a Source Data file.

repeated charge/discharge conditions. The above considerations suggest that KPSs migration can play an important role in KSBs, which unfortunately has been totally overlooked. This motivated us to develop sulfur host with combined functionalities of accelerating KPSs migration and catalyzing KPSs conversion for pursuing high-performance KSBs.

In this study, to implement the principle of optimizing KPSs behavior, the coupling of tungsten single atom ($W_{SA}$) and tungsten carbide ($W_2C$) is designed based on density functional theory (DFT) calculation. Creating different coordination environments for tungsten in metal organic framework precursor drives the symbiosis of $W_{SA}$ and $W_2C$ on nitrogen-doped carbon (NC) upon pyrolysis. Employing the obtained $W_{SA}$-$W_2C$@NC as sulfur host enables the KSBs to deliver ultra-high capacity of 1504 mAh g$^{-1}$, rate performance (1059 mAh g$^{-1}$ at 1675 mA g$^{-1}$) and stability (cycling over 200 cycles). Combined spectroscopic characterizations and theoretical computations revealed the critical role of KPSs optimization in avoiding inert sulfides

accumulation and catalytic poisoning, as well as enhancing rate and cycling performances of KSBs.

## Results

### Theoretical guidance and screening for sulfur host design

The interactions between the functional species in sulfur hosts and potassium polysulfides (KPSs) are comprehensively investigated to acquire the principles of the host design[21,36]. Regarding to the catalytic conversion functionality towards KPSs, the chemical binding of key polysulfide on the sulfur host is the primary factor[21,36,37]. For saving trial-and-error experiments, density functional theory (DFT) calculation was first conducted to screen the optimal catalytic species. The stoichiometric middle point of the whole KPSs conversion, i.e. $K_2S_4$, was employed as the representative for the binding energy calculations. Figure 1a lists the tungsten species including tungsten metal/carbides/nitride/oxide/sulfide/selenide and nitrogen coordinated single atom tungsten ($W_{SA}$) for the theoretical screening. The specific

crystal planes of tungsten species to interact with $K_2S_4$ were determined based on the maximum surfaces of the Wulff configurations (Supplementary Fig. 1 and Supplementary Table 1). The optimized binding configurations of $K_2S_4$ on tungsten species and corresponding binding energy values were displayed in Fig. 1a. $W_2C$ (102) was screened out by the highest binding energy for $K_2S_4$ (−3.01 eV).

Apart from the KPSs conversion catalysis, the facile KPSs migration over sulfur hosts should be considered with equal importance due to the mismatch between ubiquitously distributed sulfur and disjunctively distributed catalytic sites. Among aforementioned tungsten species, only $W_{SA}$ can achieve the atomically homogeneous dispersion on the host, because the compounds, even being nanostructured, can only exist as spatially discrete islands. Therefore, we calculated the migration kinetics of $K_2S_2$ (key intermediate of solid-state KPSs conversion and midpoint of $S^0/S^{2-}$ redox) on $W_2C$, $W_{SA}$ modified nitrogen-doped carbon ($W_{SA}@NC$) and NC. Figure 1b demonstrates the initial, transition and final states of $K_2S_2$ migration on NC, $W_{SA}@NC$, and $W_2C$ (102). The corresponding energy barriers for $K_2S_2$ migrations are determined to be 0.99, 0.74, and 1.30 eV. Evidently, $W_{SA}$ could effectively lower the energy barrier of $K_2S_2$ migration, enabling facile transfer of solid-state KPSs[36,38].

The comprehensive consideration of the catalytic KPSs conversion and facilitated KPSs migration results in the volcano-type relationship in Fig. 1d. At the vertex of the volcano, $W_2C$ and $W_{SA}$ should be selected as dual functional components in sulfur host. Figure 1e is the scheme demonstrating the design principle of the sulfur host guided by the theoretical calculation. $W_2C$ nanocrystals provide isolated catalytic sites for KPSs conversion, while $W_{SA}$ modified carbon substrate act as highways for fast KPSs transportation. During the discharge/charge processes, KPSs could be fed to $W_2C$ sites for catalytic conversion, and afterwards the reaction products could also be promptly released away from $W_2C$ sites. This mechanism can effectively address the distribution mismatch of sulfur and catalytic sites. Moreover, the facile solid-state KPSs migration surrounding $W_2C$ is expected to be highly conducive to the durability of the catalytic sites by avoiding catalytic poisoning.

## Fabrication and characterization of sulfur hosts and sulfur cathodes

To experimentally implement the conceived sulfur host based on theoretical guidance, we employed a sophisticated method to construct composite of tungsten single atoms and tungsten carbide nanocrystals on nitrogen-doped porous carbon (termed $W_{SA}$-$W_2C@NC$). Figure 2a demonstrates the synthesis procedure. ZIF-8 nanoparticles were prepared as nitrogen containing carbon precursor for loading tungsten. $WO_4^{2-}$ anions as tungsten sources were introduced into ZIF-8 matrix through solvothermal treatment. In the obtained $WO_4^{2-}$-ZIF product, a portion of $WO_4^{2-}$ anions incorporate into the organic ligands of zinc atoms in ZIF-8 through an anion exchange process. Meanwhile, another portion of $WO_4^{2-}$ anions adsorbed on the ZIF-8 surface by the electrostatic effect. Therefore, two different coordination environments were created for tungsten in $WO_4^{2-}$-ZIF precursor, which induced different tungsten coalescence processes during the subsequent pyrolysis. The $WO_4^{2-}$ anions coupling with the organic ligands of ZIF-8 underwent a spatial confinement pyrolysis, which transmute into nitrogen coordinated single tungsten atoms ($W_{SA}$). $WO_4^{2-}$ anions on ZIF-8 surface agglomerated and formed tungsten carbides in the carbon rich atmosphere. ZIF-8 matrix was carbonized into nitrogen-doped porous carbon, supporting the $W_{SA}$ and $W_2C$ species.

The obtained $W_{SA}$-$W_2C@NC$ inherited the polyhedron morphology (Supplementary Fig. 2). Aberration-corrected high angle annular dark-field scanning transmission electron microscopy (HAADF-STEM) and high-resolution transmission electron microscopy (HRTEM) images (Fig. 2b, Supplementary Figs. 3 and 4a) demonstrated that

tungsten species in $W_{SA}$-$W_2C@NC$ include the $W_2C$ nanocrystals (dashed circles) and single tungsten atoms (solid circles). Single tungsten atoms distribute over the whole NC substrate and embrace $W_2C$ nanocrystals. Of note, the small-size and relatively low loading $W_2C$ nanocrystals cannot be detected by XRD (Supplementary Fig. 5). For determining the optimal content of $W_{SA}$-$W_2C$ in the hosts, specimens with higher $W_{SA}$-$W_2C$ content ($W_{SA}$-$W_2C$-H@NC) and lower $W_{SA}$-$W_2C$ content ($W_{SA}$-$W_2C$-L@NC) were also prepared. The $W_{SA}$ and $W_2C$ species in these two control specimens could be well identified by SEM, HRTEM and HAADF-STEM images (Supplementary Fig. 6).

$W_2C@NC$ was prepared by eliminating the solvothermal anion exchange process during synthesis, in which case only electrostatically adsorbed $WO_4^{2-}$ existed in the precursor. In HRTEM and HAADF-STEM images (Supplementary Figs. 4b and 7), only tungsten carbide ($W_2C$) nanocrystals were present with scarcely any single tungsten atom observed. Similar to $W_{SA}$-$W_2C@NC$, the $W_2C$ nanocrystals are too small to be detected by XRD (Supplementary Fig. 5). This result verifies the importance of creating different coordination environments of tungsten in the precursor for achieving the symbiosis of $W_{SA}$ and $W_2C$ upon pyrolysis. Baseline of nitrogen-doped porous carbon (NC) was prepared by directly carbonizing tungsten-free ZIF-8. HRTEM images and selected area electron diffraction (SAED) patterns confirmed the amorphous tissue of NC (Supplementary Fig. 8).

The electronic structures and atomic configurations of tungsten species in $W_{SA}$-$W_2C@NC$ were verified by X-ray absorption near-edge structure (XANES) and Fourier transform extended X-ray absorption fine structure (EXAFS) spectra. In Fig. 2c, the white line peak in the W $L_3$-edge XANES spectrum of $W_{SA}$-$W_2C@NC$ lies between tungsten metal foil and $WO_2$, indicating an average valence state between 0 and +4. EXAFS spectrum demonstrates the distinct W–N peak at ~1.35 Å and W–C peak at ~2.67 Å[39–41], which verifies the coexistence of single tungsten atoms and tungsten carbides in $W_{SA}$-$W_2C@NC$.

Employing the aforesaid hosts, $W_{SA}$-$W_2C@NC/S$, $W_2C@NC/S$, and NC/S were prepared by melting-impregnation method. The products retained the morphologies of the pristine hosts (Supplementary Fig. 9). No isolated sulfur particle could be observed, suggesting the full infiltration of sulfur in the hosts. According to HRTEM images, the size and crystal structures of the tungsten carbides remain unchanged after sulfur loading for both $W_{SA}$-$W_2C@NC/S$ (Fig. 2e) and $W_2C@NC/S$ (Supplementary Fig. 10). Control specimens of $W_{SA}$-$W_2C$-H@NC/S and $W_{SA}$-$W_2C$-L@NC/S display the same phenomenon, as revealed in Supplementary Fig. 11. X-ray photoelectron spectroscopy (XPS) demonstrated the interaction between sulfur and the host[28,42,43]. In the W $4f$ spectra (Fig. 2f), the sulfur loading induced an elevated proportion of $W^{5+}$ and an increase in average valence from +3.43 to +4.07 for $W_{SA}$-$W_2C@NC/S$. By contrast, the W $4f$ in $W_2C@NC/S$ hardly changed (Supplementary Fig. 12). This phenomenon indicates that the interaction between sulfur and atomic-level homogeneously dispersed $W_{SA}$ is more pronounced, which also suggests the sulfur distribution all over the entire host.

The HAADF-STEM images and elemental mappings in Fig. 2g and Supplementary Fig. 13 revealed that sulfur uniformly distributed in $W_{SA}$-$W_2C@NC$, $W_2C@NC$ and NC hosts. SAED patterns (Supplementary Fig. 14) display no diffraction spots or rings of crystalline sulfur, suggesting the amorphous texture of sulfur in these specimens. The absence of cyclo-$S_8$ characteristic peaks in the Raman spectra (Supplementary Fig. 15) agreed with the SAED data[22]. According to the nitrogen adsorption-desorption isotherms analyses (Supplementary Fig. 16), the significantly decreased surface areas of the composites as compared to pristine hosts (e.g. 15.96 m² g⁻¹ of $W_{SA}$-$W_2C@NC/S$ versus 706.60 m² g⁻¹ of $W_{SA}$-$W_2C@NC$) confirmed the effective sulfur infusion. Moreover, the sulfur hosts all provide microporosities for sulfur accommodation, which should be the essential reason for the amorphous structure of the impregnated sulfur[44].

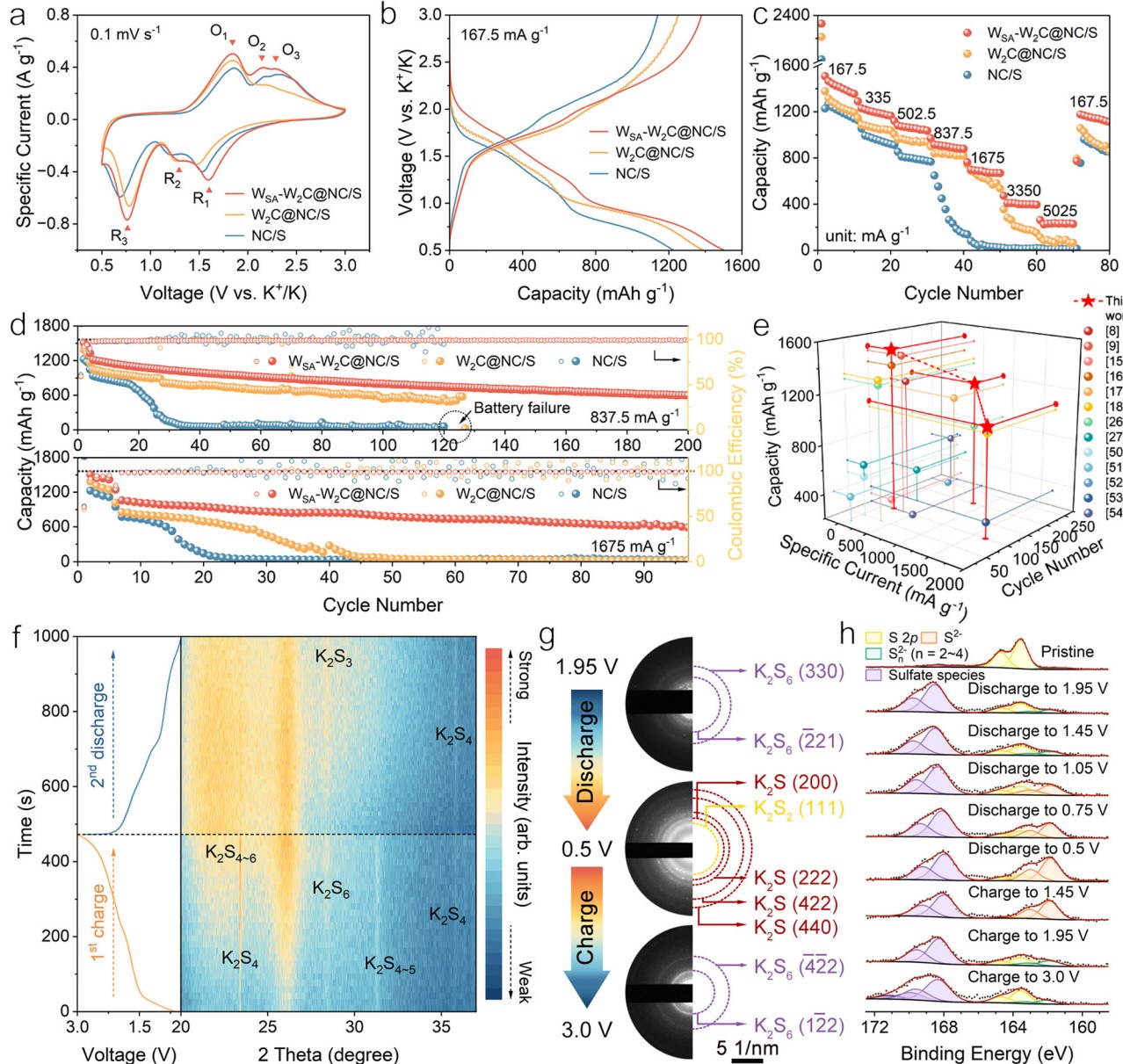

**Fig. 3 | Electrochemical performances and sulfur redox mechanism. a** CV curves of KSBs employing $W_{SA}$-$W_2C@NC/S$, $W_2C@NC/S$, and NC/S cathodes. **b–d** GCD profiles curves at 167.5 mA g$^{-1}$ (**b**) rate-performance (**c**) and cyclability (**d**) of KSBs employing three cathodes. **e** Comparison of specific capacity, rate and cycling performance of $W_{SA}$-$W_2C@NC/S$ cathode with the state-of-the-art cathodes for KSBs reported. **f** In-situ XRD of $W_{SA}$-$W_2C@NC/S$ cathode in the 1st charge and 2nd discharge processes. **g** Ex-situ SAED patterns of $W_{SA}$-$W_2C@NC/S$ cathode at various discharge/charge states. **h** Ex-situ S 2p XPS spectra of $W_{SA}$-$W_2C@NC/S$ cathode at selected discharge/charge states. Source data are provided as a Source Data file.

## Electrochemical performances and sulfur redox mechanisms

$W_{SA}$-$W_2C@NC/S$ utilizing the host containing functional species of $W_{SA}$ and $W_2C$ is expected to be high-performance cathodes for KSBs. First, $W_{SA}$-$W_2C@NC$ and $W_2C@NC$ demonstrated much stronger adsorption capability towards KPSs than that of tungsten-free NC, as proved by the ultraviolet-visible (UV-vis) spectra of $K_2S_6$ solutions with different host materials (Supplementary Fig. 17)[45,46]. This phenomenon agrees well with the calculation demonstrated in Fig. 1. After 12 h, all $K_2S_6$ species in the solution were thoroughly adsorbed by $W_{SA}$-$W_2C@NC$ and $W_2C@NC$ hosts, as evidenced by the corresponding spectra overlapping with pure DME solvent. To understand the sulfur redox in $W_{SA}$-$W_2C@NC/S$, $W_2C@NC/S$ and NC/S cathodes, cyclic voltammetry (CV) measurements were conducted. As shown in Supplementary Fig. 18, in the first cathodic scan of $W_{SA}$-$W_2C@NC/S$, there is a strong peak at 0.836 V combining the SEI formation[9,19,22,32] and sulfur

activation processes[8,47–49]. The sulfur activation describes the potassiation of pristine sulfur, which probably needs to overcome higher energy barrier than the following potassiation processes[22,44,49]. Therefore, the cathodic peak of the sulfur potassiation in the 1st cycle CV appeared at lower onset voltage than that in the following cycles[9,19,22]. Reduction peaks at 1.594 (R$_1$), 1.329 (R$_2$), and 0.761 V (R$_3$) appeared in the 2nd cycle (Fig. 3a). In the anodic scan, oxidation peaks present at 1.844 (O$_1$), 2.159 (O$_2$), and 2.272 V (O$_3$). $W_2C@NC/S$ and NC/S exhibit similar CV shapes as $W_{SA}$-$W_2C@NC/S$, but deliver lower response specific currents and larger polarizations. This phenomenon suggests the faster sulfur redox kinetics for $W_{SA}$-$W_2C@NC/S$ than the other two cathodes.

The galvanostatic charge/discharge (GCD) curves of the three cathodes (2nd cycle) were displayed in Fig. 3b. The $W_{SA}$-$W_2C@NC/S$ cathode delivered a high discharge capacity of 1504 mAh g$^{-1}$ based on

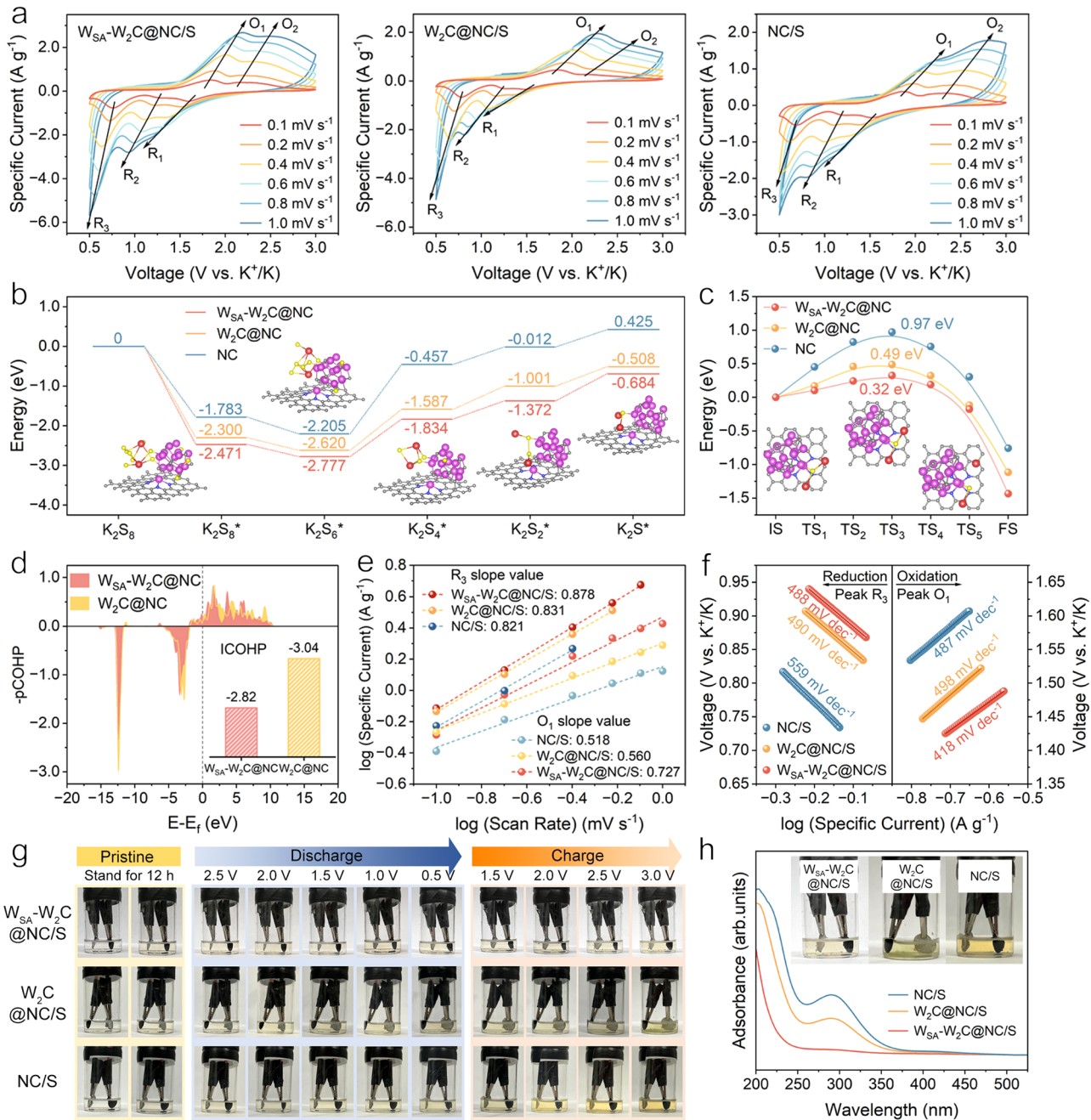

**Fig. 4 | Sulfur redox kinetics under synergy strategy. a** Multi-rate scan CV curves of KSBs employing $W_{SA}$-$W_2C@NC/S$, $W_2C@NC/S$, and NC/S cathodes. **b** Gibbs free energy profiles of sulfur reduction processes on NC, $W_2C@NC$, and $W_{SA}$-$W_2C@NC$ hosts. **c** Energy profiles for $K_2S$ dissociation on $W_{SA}$-$W_2C@NC$, $W_2C@NC$ and NC. **d** Projected crystal orbital Hamilton population (pCOHP) diagrams of W–S bonds linking $K_2S$ and $W_{SA}$-$W_2C@NC$, $W_2C@NC$ (insert shows ICOHP values). **e** Linear relationship of log (specific current) vs. log (scan rate) of peaks $R_3$ and $O_1$ for these three cathodes. **f** Tafel slope of peaks $R_3$ and $O_1$ derived from CV curves at 0.2 mV s$^{-1}$. **g** Visual cells under different states employing $W_{SA}$-$W_2C@NC/S$, $W_2C@NC/S$, NC/S cathodes and potassium foil anodes. **h** UV-vis spectra and the optical photos (inset) of visual cells employing different cathodes at the end of charge process. Source data are provided as a Source Data file.

the thermogravimetric analysis (TGA) determined sulfur content (40.56%) (Supplementary Fig. 19). This reversible capacity corresponds to 89.8% sulfur utilization, which is the highest value reported by far. The counterpart values for $W_2C@NC/S$ and NC/S are 82.8% and 73.3%, respectively, highlighting the advantage of $W_{SA}$-$W_2C$ composite in enhancing the electrochemical activity of sulfur. The discrepancy in sulfur content among three cathodes is below 3.4%, which is at the lowest level in the field (Supplementary Table 2). The cathodes with such sulfur content discrepancy deliver negligible difference in electrochemistry (Supplementary Fig. 20). The absolute sulfur contents above 40% are higher than the reported sulfur-microporous carbon

composites and on par with SPANs (Supplementary Table 3). The extremely low capacities of the pure hosts (Supplementary Fig. 21) verified that the capacities of cathodes are provided by the reversible sulfur redox. Of note, the three specimens exhibited highly similar nitrogen moieties in terms of nitrogen types and relative ratios (Supplementary Fig. 22 and Supplementary Table 4). Therefore, the different electrochemical performances of three cathodes should be essentially stemmed from the tungsten species. $W_{SA}$-$W_2C@NC/S$ electrodes with high sulfur loading of 2.86 and 3.61 mg cm$^{-2}$ delivered slightly lower capacities of 1477.8 and 1221.9 mAh g$^{-1}$ (Supplementary Fig. 23), which are higher than most previously reported cathodes with

low sulfur loading (Supplementary Table 3). Control cathodes of $W_{SA}$-$W_2C$-H@NC/S and $W_{SA}$-$W_2C$-L@NC/S delivered capacities of 1274.4 and 1416.1 mAh g$^{-1}$ (Supplementary Fig. 24), suggesting $W_{SA}$-$W_2C$@NC/S has the optimal $W_{SA}$-$W_2C$ content.

Regarding to the rate capability (Fig. 3c), $W_{SA}$-$W_2C$@NC/S cathode exhibited capacities of 914, 697, 408, and 223 mAh g$^{-1}$ at current densities of 838, 1675, 3350, 5025 mA g$^{-1}$, respectively. $W_2C$@NC/S cathodes delivered lower capacities than $W_{SA}$-$W_2C$@NC/S at all rates. It is worth special noting that the capacity fast decayed for $W_2C$@NC/S at 3350–5025 mA g$^{-1}$, resulting in cell failure at high rate. This issue was more severe for NC/S cathode, as indicated by the rapid cell failure at 837.5 mA g$^{-1}$. Considering the different constitutions of tungsten species for $W_{SA}$-$W_2C$@NC and $W_2C$@NC, it is reasonable to speculate that the cooperation between $W_{SA}$ and $W_2C$ plays a critical role in maintaining the cell operation because more facile sulfur redox kinetics are required at high specific currents.

The cycling performance were also measured in addition to the rate tests. As shown in Fig. 3d, after 5 activation cycles at 167.5 mA g$^{-1}$, $W_{SA}$-$W_2C$@NC/S delivered capacities of 1214 and 1059 mAh g$^{-1}$ at 837.5 and 1675 mA g$^{-1}$, respectively. Afterwards, the cathodes kept stable cycling for 200 and 97 cycles, resulting in capacity retention ratio of 49.4 and 56.8%. $W_{SA}$-$W_2C$-H@NC/S and $W_{SA}$-$W_2C$-L@NC/S showed inferior cyclability, again proving the optimal content of $W_{SA}$-$W_2C$ in $W_{SA}$-$W_2C$@NC (Supplementary Fig. 24). For comparison, $W_2C$@NC/S cathode can last for only 127 and 40 cycles at 837.5 and 1675 mA g$^{-1}$. The cell failure occurred even earlier for NC/S cathode, i.e. at 32$^{nd}$ and 20$^{th}$ cycles. The cell failure can only be postponed by cycling the cathodes at very slow rate (Supplementary Fig. 25). The changes in resistances in the cells provide another aspect reflecting the diversity in cyclability of the cathodes (details in Supplementary Fig. 26 and Table 5). Combining the rate and cycling performance, it could be concluded that certain injurious process continuously deteriorated sulfur redox property and finally damaged the cell, which can be aggravated at higher specific currents. More importantly, the synergy of the functionalities of $W_{SA}$ and $W_2C$ can significantly alleviate this negative factor.

It is instructive to compare $W_{SA}$-$W_2C$@NC/S based KSB with the state-of-the-art KSBs reported. As shown in Fig. 3e and Supplementary Table 3, KSBs employing SPAN cathodes[15–17], small-molecular sulfur cathodes[8,9,50], polysulfide catholyte[51] and cyclo-$S_8$ cathodes[18,26,27,52–54] are all included for comprehensive comparison. Remarkably, $W_{SA}$-$W_2C$@NC/S cathode reaches the highest sulfur utilization and best rate performance. It is worth special noting that the cyclability of $W_{SA}$-$W_2C$@NC/S cathode is comparable to that of most robust SPAN cathodes, suggesting the advantages of $W_{SA}$-$W_2C$@NC/S for practical KSBs.

To pursue insights of the K-S redox mechanism, in-situ XRD, ex-situ SAED and XPS were conducted to investigate the $W_{SA}$-$W_2C$@NC/S cathode during charge and discharge. Per Fig. 3f, as the charging process proceeded, the peaks of $K_2S_6$ emerged and gradually strengthened at the expense of $K_2S_4$ peak. In the following discharge process, the decrease of $K_2S_{4-6}$ peak intensity and increase of $K_2S_3$ peak intensity can be observed, suggesting the reversible conversion between $K_2S_6$ and $K_2S_3$ in $W_{SA}$-$W_2C$@NC/S cathode. The diffraction signals of $K_2S_2$ and $K_2S$ cannot be easily distinguished in the in-situ XRD patterns probably due to the short time for collecting each XRD pattern and the small size of target phases. Thus, SAED was used to further identify the reaction products at various discharge/charge states (Supplementary Fig. 27). Starting from the amorphous halo for pristine cathode (Supplementary Fig. 28), a bright diffraction ring corresponding to $K_2S_6$ appeared upon discharge (Fig. 3g). As the discharge continued, $K_2S_6$ ring gradually diminished (Dis-1.45 V). Diffraction rings ascribed to $K_2S$ phase appeared and became stronger (Dis-1.05/0.75 V). At the lowest discharge voltage of 0.5 V, only strong diffraction rings of $K_2S$ and relatively weak diffraction ring of $K_2S_2$ existed,

suggesting the $K_2S$ and minor $K_2S_2$ as the discharge end products. During the reversible charge process, diffraction rings ascribed to $K_2S_4$ first appeared (Cha-1.45 V). Upon further charging, $K_2S_6$ reappeared (Cha-1.95/3.0 V), suggesting the fully reversible conversion from $K_2S$ to $K_2S_6$ in $W_{SA}$-$W_2C$@NC/S cathode.

Ex-situ XPS characterizations revealing valence change of sulfur were also conducted as supplement analyses on KPSs conversion. In Fig. 3h, a clear increase of $S_n^{2-}$ (163.4 eV, 162.2 eV) and $S^{2-}$ (163.0 eV, 161.8 eV) components at the expense of $S^0$ (164.7 eV, 163.5 eV) can be observed in discharge[55,56]. $S^{2-}$ started to convert to $S_n^{2-}$ as charge back to 1.95 V. At the cut-off charge voltage of 3.0 V, both $S_n^{2-}$ and $S^{2-}$ completely converted to $S^0$. The XPS observations are highly consistent with the analyses derived from in-situ XRD and ex-situ SAED patterns.

The sulfur redox process can be resolved based on the systematic spectroscopic data. With regard to the CV curve in Fig. 3a, $R_1/R_2$ should be ascribed to the successive conversion of long-chain to short-chain KPSs, and $R_3$ is associated with the reduction of short-chain KPSs into $K_2S$. In the anodic scan, $O_3/O_2$, and $O_1$ should be ascribed to the long-chain KPSs oxidation and solid-state KPSs oxidation, respectively. Comparing the peak intensities in the CV curves, the prominent long-chain KPSs conversion for NC/S, and facile short-chain KPSs conversion for $W_2C$@NC/S can be well distinguished. Meanwhile, $W_{SA}$-$W_2C$@NC/S cathode exhibited the most facile kinetics in all the steps of long-/short-chain KPSs conversion.

## Sulfur redox kinetics under synergy of KPS conversion and migration

Based on the index of the redox peaks on CV, more in-depth kinetic analysis of the cathodes can be conducted according to multi-rate CV scanning. The different trends of response specific currents and voltage polarizations as a function of scan rate reflect the kinetics of specific sub-steps of sulfur redox (Fig. 4a). First, the comparison of $R_1/R_2$ specific currents and polarizations for $W_2C$@NC/S and NC/S demonstrated that $W_2C$ catalytic site had no significant promotion on the long-chain KPSs conversion. $W_{SA}$-$W_2C$@NC/S cathode delivered ca. 25% higher $R_1$ specific current, which may probably be attributed to the facilitated long-chain KPSs migration across the carbon matrix by $W_{SA}$ moieties. The most facile long-chain KPSs reduction kinetics enabled by $W_{SA}$-$W_2C$ can also be demonstrated by the CV profiles of $K_2S_6$ based symmetric cell (Supplementary Fig. 29)[57–59].

Regarding to the $R_3$ peak ascribed to the solid-state KPSs reduction, $W_{SA}$-$W_2C$@NC/S displayed the lowest polarization and highest response specific current, suggesting the most facile kinetics of solid-state $K_2S_4$-$K_2S$ conversion. $W_2C$@NC/S displays lower $R_3$ specific current than that of $W_{SA}$-$W_2C$@NC/S, revealing the synergy of $W_{SA}$ and $W_2C$ in enhancing solid-state reaction kinetics. Comparing the characteristics of $R_3$ for $W_2C$@NC/S and NC/S, $W_2C$@NC/S exhibited an accelerated $R_3$ redox, as indicated by the larger response specific currents than that of NC/S. The $R_3$ peak for NC/S is much weaker especially at high scan rate suggesting the difficulty of solid-state KPSs conversion to reach $K_2S_2/K_2S$ without the catalytic effect of $W_2C$ nanocrystals.

To reveal the origin of the different sulfur reduction kinetics for the three cathodes, the Gibbs free energy changes along the sulfur reduction pathways were calculated. Figure 4b demonstrates the thermodynamically stable configurations of various KPSs binding with different hosts and the corresponding free energy profiles. It can be distinguished that the reaction sub-steps from $K_2S_8$ to $K_2S_6$ are exothermic for three cathodes. The successive sub-steps of reduction (from $K_2S_6$ to $K_2S$) are all endothermic. The maximum energy barriers occur at the steps of $K_2S_6$ to $K_2S_4$ for all three cathodes, which are identified as the rate-determining step (RDS) of the whole reduction process. The highest RDS energy barrier (1.748 eV) for NC/S explains the difficulty of the long-chain to short-chain KPSs conversion. With the catalytic effect of $W_2C$ nanocrystals, the lower RDS energy barrier

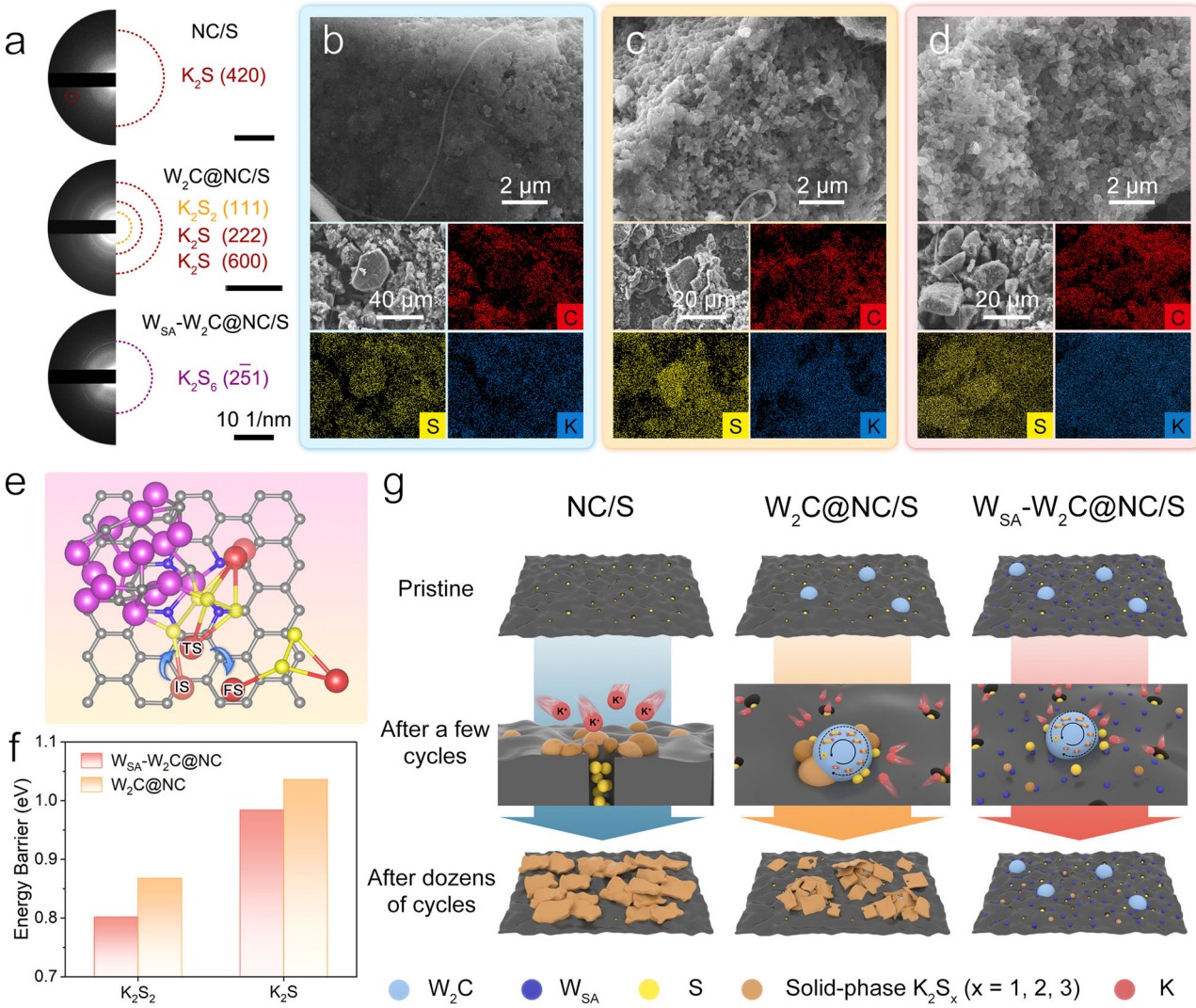

**Fig. 5 | Investigations on sulfur redox durability in KSBs. a** SAED patterns of post-cycled NC/S, W$_2$C@NC/S, or W$_{SA}$-W$_2$C@NC/S cathodes. **b–d** SEM images and corresponding EDS mapping of post-cycled NC/S (**b**), W$_2$C@NC/S (**c**) and W$_{SA}$-W$_2$C@NC/S (**d**) cathodes. **e** Configuration evolution of K$_2$S$_2$ migration from W$_2$C catalytic site to W$_{SA}$ modified NC substrate. **f** Energy barriers of K$_2$S$_2$ and K$_2$S migration from W$_2$C catalytic site to W$_{SA}$ modified NC substrate or pure W$_{SA}$-free NC substrate. **g** Schematic illustration of the potassium-sulfur electrochemistry in KSBs employing NC/S, W$_2$C@NC/S and W$_{SA}$-W$_2$C@NC/S cathodes. Source data are provided as a Source Data file.

(1.033 eV) endows W$_2$C@NC/S with more pronounced solid-state KPSs reduction. As for W$_{SA}$-W$_2$C@NC/S, the lowest RDS energy barrier (0.943 eV) ensured the advantage in kinetics of both long-chain and short-chain conversion. This optimally balanced multi-step successive reduction process was consistent with the prominent peaks of both R$_{1,2}$ and R$_3$ displayed in the CV curves.

Regarding to the anodic scan, peaks of O$_1$ and O$_2$ can be distinguished for all three cathodes, which are ascribed to solid-state KPSs oxidation and long-chain KPSs oxidation, respectively. Of note, the O$_3$ peaks distinguishable in the low scan rate (Fig. 3a) integrated into the O$_2$ peak as scan rate increased, thereby being considered as one characteristic O$_2$ peak. The response specific current of O$_2$ peak for NC/S enhanced with increasing scan rate, suggesting the intrinsically facile oxidation of long-chain KPSs. The O$_1$ peak is much lower than O$_2$ peak for NC/S at high scan rate, which is different from that for W$_{SA}$-W$_2$C@NC/S and W$_2$C@NC/S. This phenomenon indicates the fewer KPSs involving in solid-state conversion during charge for NC/S due to the absence of catalytic species. In the cases of W$_2$C@NC/S, the O$_1$ peak became more pronounced, suggesting the catalytic effect of W$_2$C towards short-chain KPSs oxidation. W$_{SA}$-W$_2$C@NC/S delivered

the highest specific currents for O$_{1,2}$ peaks, highlighting the most favorable oxidation process from K$_2$S to long-chain KPSs. The K$_2$S$_6$ based symmetric cell tests further proved the most facile long-chain oxidation for W$_{SA}$-W$_2$C@NC (Supplementary Fig. 29).

The K$_2$S decomposition is the initial step of charging, which is also the most critical obstacle for efficient K-S electrochemistry in KSBs[18,26,27]. Therefore, the decomposition energy barriers of K$_2$S on W$_{SA}$-W$_2$C@NC, W$_2$C@NC and NC substrates were calculated to identify the reason for different charging behaviors of the cathodes (Fig. 4c). Based on the configuration determination of initial, transition and final states, the energy barriers for K$_2$S decomposition are respectively 0.32, 0.49 and 0.97 eV for W$_{SA}$-W$_2$C@NC, W$_2$C@NC and NC, suggesting the highest catalytic activity of W$_{SA}$-W$_2$C site towards K$_2$S decomposition.

As the transition state provides a snapshot of the real-time K$_2$S dissociation process, the characteristic of transition configuration reveals the origin of catalytic activity (Supplementary Fig. 30). The strength of W–S bond between W$_2$C and K$_2$S derives from the *d-p* orbital hybridization[37,60], which is an effective descriptor for the catalytic capability[18,35]. Basically, weaker W–S bond strength would lead to

more facile K$_2$S decomposition and lower overpotential of K$_2$S oxidation[18]. Projected crystal orbital Hamilton population (pCOHP) analysis were performed to evaluate the W–S bond strength in the transition states of dynamic interaction between K$_2$S and W$_{SA}$-W$_2$C, W$_2$C sites (Fig. 4d). According to the integrated-COHP (ICOHP) values, the W–S bond between K$_2$S and W$_{SA}$-W$_2$C site is weaker due to the modulation of W$_{SA}$ moieties[37,61,62], suggesting the easier K$_2$S dissociation stemming from the catalytic effect of W$_{SA}$-W$_2$C.

To experimentally evaluate the effect of different tungsten species towards the sulfur redox kinetics, the Tafel slopes and b values were extracted from multi-rate scan CV curves. The b value in $i = a\,v^b$ reflects the reaction kinetics ($i$ and $v$ are the response specific current and scan rate)[63,64]. As shown in Fig. 4e, the b values of R$_3$ (O$_1$) peaks for W$_{SA}$-W$_2$C@NC/S, W$_2$C@NC/S and NC/S were 0.878 (0.727), 0.831 (0.560) and 0.821 (0.518), respectively. The highest b values of R$_3$/O$_1$ for W$_{SA}$-W$_2$C@NC/S cathode indicate the most facile solid-state KPSs conversion, which is the essential reason for the highest sulfur utilization, optimal rate and cycling capability.

The redox kinetics associated with short-chain KPSs conversion can also be interpreted by the Tafel slopes. In Fig. 4f, both W$_{SA}$-W$_2$C@NC/S and W$_2$C@NC/S exhibited earlier R$_3$ onset potential and distinctly lower Tafel slope than NC/S (488, 490 vs. 559 mV dec$^{-1}$), which is directly related to the accelerated reduction of solid KPSs by the tungsten species catalysis. In the initial oxidation process, W$_{SA}$-W$_2$C@NC/S provided a more positive onset potential and exhibited a lower Tafel slope of 418 mV dec$^{-1}$ for the decomposition of solid short-chain KPSs (K$_2$S$_2$/K$_2$S) compared to W$_2$C@NC/S and NC/S.

The inferior Tafel slope of W$_2$C@NC/S reveals the subtle difference in reaction mechanism caused by the participation of W$_{SA}$. Due to the lower energy barrier for K$_2$S$_2$ migration on W$_{SA}$ modified NC than that on W$_{SA}$-free NC, the presence of W$_{SA}$ facilitates the mass transfer around the W$_2$C catalytic center, thereby maximizing the overall efficiency of K$_2$S oxidation. Instead, the sluggish transportation of oxidation product could cause accumulation of insulting solid KPSs (e.g. K$_2$S$_2$) around the catalytic sites diminishing the activity for successive reaction (named catalytic poisoning)[65]. The W$_{SA}$-W$_2$C composite synergizing the catalytic conversion and facilitated migration towards KPSs is the key for the highly efficient solid-state KPSs conversion. Of note, W$_2$C@NC/S and NC/S delivered comparable Tafel slopes near the O$_1$ peaks, which is direct evidence of the poisoning effect on W$_2$C catalytic site in W$_2$C@NC/S.

The more facile KPSs conversion kinetics endowed by the synergy of W$_{SA}$ and W$_2$C can also be demonstrated by visual cell measurements. As shown in Fig. 4g, the visual cells employing potassium foils as anodes and W$_{SA}$-W$_2$C@NC/S, W$_2$C@NC/S, NC/S as cathodes underwent 12 h standing and one entire discharging/charging cycle. The amount of KPSs dissolving in the electrolyte during this process is a reasonable descriptor for the KPSs conversion kinetics, because sluggish KPSs conversion provides more time for KPS dissolving. At the end point of charge-3.0 V, the electrolyte for W$_{SA}$-W$_2$C@NC/S visual cell remained colorless with negligible KPS signal detected in UV-vis spectrum (Fig. 4h). On the contrary, the electrolytes for W$_2$C@NC/S and NC/S visual cells dissolved much more KPS, suggesting the less facile KPS conversion.

### Sulfur redox durability under synergy of KPS conversion and migration

The analyses above revealed the decisive role of W$_{SA}$-W$_2$C composite in affecting the kinetics of solid-state KPSs conversion. To demonstrate the correlation between the solid-state KPSs conversion and cyclability of KSBs, the post-cycled cathodes (in charged states) were collected and examined. As shown in Fig. 5a and Supplementary Fig. 31, the majority phases of sulfur species are determined to be K$_2$S for NC/S, K$_2$S for W$_2$C@NC/S, and K$_2$S$_6$ for W$_{SA}$-W$_2$C@NC/S. The result reveals that the sulfur species in NC/S and W$_2$C@NC/S became mainly inert

K$_2$S after cycling, which cannot be oxidized back to provide reversible capacity.

The morphology and elemental distribution of the post-cycled cathodes were demonstrated in Fig. 5b-d. Severe agglomeration occurred in NC/S cathode, and the original polyhedrons can no longer be discerned, which is due to the extremely thick accumulation of sulfide cementing the polyhedrons[66,67]. The element mapping of sulfur, potassium and carbon reveals the large size (ca. 30 μm) potassium sulfide particles that detached from the carbon matrix. This phenomenon also appeared for post-cycled W$_2$C@NC/S cathode (Fig. 5c). Although the polyhedron agglomeration is less severe, the non-uniform sulfur and potassium distribution reveal the segregation of potassium-rich sulfides in the cathode. As for post-cycled W$_{SA}$-W$_2$C@NC/S cathode, the polyhedron morphology almost remained unchanged (Fig. 5d). More importantly, the sulfur mapping overlapped with the shape of micro-size particles, but the potassium distributes uniformly all over the cathode, suggesting the absence of potassium-rich sulfide segregation.

The SEM/EDS results supplement the SAED analyses, further verifying that the accumulation of inert potassium sulfides is the essential reason causing the failure of W$_2$C@NC/S and NC/S based KSBs. We claim that the facilitated migration of the potassium sulfide by W$_{SA}$ plays a key role in significantly alleviating the sulfide accumulation. To prove this point, the migration behaviors of K$_2$S$_2$ and K$_2$S from W$_2$C catalytic site to the carbon substrate were investigated (Fig. 5e and Supplementary Fig. 32). The corresponding energy barriers for the migrations were summarized in Fig. 5f. Apparently, with the participation of W$_{SA}$, the transfers of K$_2$S$_2$ and K$_2$S from catalytic site can be accelerated due to the distinctly lower migration energy barrier. During the discharge, the fast detachment of K$_2$S reduction product from catalytic site avoids the local accumulation of insulating K$_2$S. In the charge, the oxidation product of K$_2$S$_2$ can efficiently diffuse from catalytic site to the substrate, alleviating the catalytic poisoning and maintaining the consecutive K$_2$S decomposition. These effects become more crucial at high specific current densities and prolonged cycling, which endows W$_{SA}$-W$_2$C@NC with superior rate capability and long cycling stability.

## Discussion

The mechanisms of potassium-sulfur electrochemistry in the cathodes are demonstrated in Fig. 5g. Due to the absence of functional species in NC host, the sulfur redox kinetics is inferior especially for the solid-state KPSs conversion. Consequently, the confined sulfur species rapidly escape from the microporosity and formed a non-uniform deposition of inert K$_2$S on host surface[68,69]. Upon repeated cycles, the K$_2$S deposition would block the pores and cause thorough passivation. The continuous consumption of electrochemically active sulfur species leads to the fast failure of KSB, which occurs earlier at high rates. Although possessing W$_2$C catalytic sites for W$_2$C@NC host, the sluggish migration of solid KPSs like K$_2$S/K$_2$S$_2$ would inevitably cause the catalytic poisoning due to the inefficiently prompt detachment of solid sulfides from the catalytic sites. In addition, the high energy barrier of KPSs migration on carbon substrate would restrict the contact between sulfur species and catalytic sites. Therefore, the accumulation of inert sulfides would continuously aggravate the cell performance upon prolonged cycling and finally lead to KSB failure[12,28]. With regard to W$_{SA}$-W$_2$C@NC/S cathode, the W$_{SA}$-W$_2$C composite synergizes the facilitated migration and catalytic conversion towards polysulfides. The facile migration of KPSs on W$_{SA}$ modified carbon substrate addresses the mismatch in distribution between ubiquitous sulfur species and isolated catalytic sites. The accelerated transfer of solid K$_2$S/K$_2$S$_2$ around W$_2$C sites prevents the accumulation of inert sulfides and alleviates the catalytic poisoning. Resultantly, high sulfur utilization can be achieved and efficient sulfur redox can be maintained at the conditions of high rates and prolonged cycling, enabling the

unprecedented reversible capacity, superior rate performance and long lifespan of KSBs.

In summary, we adopted the strategy of synergizing the facilitated KPSs migration and catalytic KPSs conversion for KSBs. Based on the theoretical screening and sophisticated synthesis procedure, NC equipped with $W_{SA}$-$W_2C$ composite was fabricated as sulfur hosts. Comprehensive experimental analyses and theoretical calculations revealed that $W_{SA}$ species optimize the catalytic capability of $W_2C$ species for solid-state KPSs conversion (more facile $K_2S$ dissociation) and accelerate the KPSs migration near catalytic sites (lower energy barrier for $K_2S/K_2S_2$ diffusion). As a result, the insulating sulfides accumulation and catalytic poisoning were effectively alleviated in $W_{SA}$-$W_2C$@NC/S cathode than the $W_{SA}$ free $W_2C$@NC/S and tungsten free NC/S counterparts, which contributes to the ultra-high sulfur utilization, efficient sulfur redox at high rate and long cycling stability. The validated strategy of KPSs migration acceleration and conversion catalysis bi-functional sulfur host brings more possibility for developing high performance KSBs.

## Methods
### Chemicals
Sodium tungstate dihydrate ($Na_2WO_4 \cdot 2H_2O$, 99%) and 2-Methylimidazole (2-MIM, 98%), zinc acetate ($Zn(CH_3COO)_2 \cdot 2H_2O$, 99.98%) were purchased from HEOWNS. Methanol (99.9%) was purchased from Shanghai Macklin Biochemical Co., Ltd. N,N-Dimethylformamide (DMF, 99.5%) was purchased from Tianjin Kemiou Chemical Reagent Co., Ltd. Hydrochloric acid (HCl, 36.0-38.0 wt.%) was purchased from Rionion Development Co., Ltd. Sulfur was purchased from Sigma-Aldrich. 0.8 M $KPF_6$ in EC/DEC was purchased from Suzhou DoDoChem Co., Ltd. N-methyl−2-pyrrolidone (NMP) and poly(vinylidene fluoride) (PVDF) were purchased from Guangdong Canrd New Energy Technology Co., Ltd. Multiwalled carbon nanotubes (MWCNTs) was purchased from Nanjing XFNANO Materials Tech Co., Ltd. Potassium metal was purchased from Alfa Aesar (China) Chemicals Co., Ltd.

### Synthesis of ZIF-8 and $WO_4^{2-}$ modified ZIF-8 ($WO_4^{2-}$-ZIF-8)
In a typical procedure, 8.95 g (0.109 mol) 2 MIM and 2.39 g (10.9 mmol) $Zn(CH_3COO)_2 \cdot 2H_2O$ were dissolved in 40 ml of deionized water under continuous magnetic stirring and labeled as solution A and solution B, respectively. Subsequently, solution B was rapidly poured into solution A and aged at room temperature for 48 h. The white ZIF-8 powder was collected by centrifugation, washed three times with methanol, and then dried overnight in a vacuum oven at 60 °C. 0.40 g dried ZIF-8 powder was dispersed in 60 ml DMF solution, and 0.20 g (0.606 mmol) $Na_2WO_4 \cdot 2H_2O$ was added afterwards. The solution was kept stirring until became homogeneous. The mixed solution was then transferred to a 100 ml Teflon-lined stainless-steel autoclave and underwent a solvothermal process at 150 °C for 12 h. After cooling to room temperature, the light yellow $WO_4^{2-}$-ZIF-8 powder was obtained by centrifugation, washing with deionized water and methanol alternately three times, and drying at 60 °C.

### Synthesis of $W_{SA}$-$W_2C$@NC
The $WO_4^{2-}$-ZIF-8 powder was heated to 400 °C and kept for 2 h under $N_2$ atmosphere with a slow heating rate of 2 °C min$^{-1}$ to prevent the formation of large compound particles during the annealing process. Subsequently, the temperature was increased to 800 °C at a rate of 2 °C min$^{-1}$ and kept for another 2 h. The obtained powders were dispersed in 50 ml 6 M HCl and held at 60 °C for 4 h to etch off the Zn species and to generate microporous. Afterward, the powder was washed with plenty of deionized water and dried in a vacuum oven at 60 °C overnight to obtain the product labeled as $W_{SA}$-$W_2C$@NC.

### Synthesis of $W_2C$@NC
0.20 g (0.606 mmol) $Na_2WO_4 \cdot 2H_2O$ and 0.40 g ZIF-8 powder was mixed and grounded in the mortar for 30 min. Afterward, the homogeneous mixture was heated up to 400 °C and kept for 2 h under $N_2$ atmosphere with a slow heating rate of 2 °C min$^{-1}$. Then, the nitrogen-filled tube furnace was ramped up to 800 °C at the same heating rate and kept for another 2 h. Afterwards, the residual Zn species in the as-prepared powder was further etched by 6 M HCl and held at 60 °C for 4 h. After collected by infiltration, the powder was washed with plenty of deionized water and dried in a vacuum oven at 60 °C overnight. The obtained product was labeled as $W_2C$@NC.

### Synthesis of NC
The ZIF-8 powder was directly carbonized according to the same two-step heating procedure. In a typical process, ZIF-8 powders were filled in a porcelain boat and heated up to 400 °C for 2 h in a nitrogen-filled tube furnace with a heating rate of 2 °C min$^{-1}$, and then to 800 °C for another 2 h. Then, the as-prepared powder was treated with 6 M hydrochloric acid solution kept at 60 °C for 4 h, followed by washing with a large amount of deionized water and then placed in a vacuum oven at 60 °C overnight. The obtained product was labeled as NC.

### Synthesis of $W_{SA}$-$W_2C$-H@NC and $W_{SA}$-$W_2C$-L@NC
The synthesis of $W_{SA}$-$W_2C$-H@NC and $W_{SA}$-$W_2C$-L@NC followed the same procedure of $W_{SA}$-$W_2C$@NC except different amount of $Na_2WO_4 \cdot 2H_2O$ (0.60 g (1.819 mmol) for $W_{SA}$-$W_2C$-H@NC and 0.10 g (0.303 mmol) for $W_{SA}$-$W_2C$-L@NC) were added for preparing $WO_4^{2-}$-ZIF-8 precursors.

### Synthesis of $W_{SA}$-$W_2C$@NC/S, $W_2C$@NC/S and NC/S composites
The as-prepared $W_{SA}$-$W_2C$@NC, $W_2C$@NC and NC hosts were uniformly mixed with sulfur powder with a mass ratio of 1:1 and sealed in a glass tube under vacuum. The glass tube was heated to 155 °C at a rate of 5 °C min$^{-1}$ and kept for 24 h, followed by heating to 300 °C and holding for 2 h. Then, the obtained powders were heated at 200 °C for 1 h under flowing Ar in a tube furnace. The products are termed as $W_{SA}$-$W_2C$@NC/S (or NC/S, $W_2C$@NC/S) composite.

### Adsorption measurements
The $K_2S_6$ solution for adsorption measurements was prepared by mixing potassium sulfide ($K_2S$) and sulfur with a molar ratio of 1:5 in dimethyl ether (DME). 5 mg $W_{SA}$-$W_2C$@NC, $W_2C$@NC, and NC were added into the 2 ml 0.02 M $K_2S_6$ solution, respectively, with the blank $K_2S_6$ solution as a reference.

### Materials characterization
The scanning electron microscopy (SEM) images were obtained by JEOL JSM-7800F with an energy dispersive spectrometer (EDS). The transmission electron microscopy (TEM) images and selected area electron diffraction (SAED) patterns were recorded by JEOL JEM-F200. The aberration-corrected high angle annular dark-field scanning transmission electron microscopy (HAADF-STEM) images were collected by JEOL JEM-ARM200F. The X-ray photoelectron spectroscopy (XPS) analysis was performed on Kratos Axis Ultra DLD spectrometer. Raman spectra were recorded with Horiba LabRam HR Evolution spectrometer. Nitrogen adsorption-desorption isotherm measurement was conducted using Micromeritics ASAP 2460 system. The thermogravimetric analysis (TGA) was investigated by using NETZSCH STA 449 C instrument. The ultraviolet-visible (UV-vis) spectra were performed by Shimadzu UV-2700 spectrophotometer. The in-situ X-ray diffraction (XRD) patterns were performed using Bruker D8 Advance with Cu Kα radiation (λ = 0.15406 nm) at 40 kV and 40 mA. The X-ray absorption structure (XAS) spectra (W L-edge) were measured in Shanghai Synchrotron Radiation Facility (SSRF).

## Electrochemical measurements

Electrochemical tests were performed by assembling CR2032-type coin cells in an Argon-filled glove box. 60 wt.% cathode materials ($W_{SA}$-$W_2C@NC/S$ (40.56 wt.% sulfur), $W_2C@NC/S$ (37.91 wt.% sulfur) or NC/S (43.93 wt.% sulfur)), 30 wt.% MWCNTs, and 10 wt.% PVDF were mixed with an appropriate amount of NMP for the preparation of a uniform slurry. The obtained slurry was cast on carbon-coated aluminum foil with the thickness of 250 μm and dried overnight in a vacuum oven at 60 °C. The electrodes were punched into 12 mm diameter disks for battery assembly. Glass fiber (GF/C, Whatman) with the diameter of 19 mm was chosen as the separator. Potassium metal foils produced from surface oxide layer-free potassium blocks were directly applied as anode. The electrolyte is 0.8 M $KPF_6$ in EC/DEC with a volume ratio of 1:1. The typical sulfur loading of the electrode is ca. 1 mg cm$^{-2}$. Electrodes with high sulfur loading of ca. 2.8 and ca. 3.6 mg cm$^{-2}$ were also prepared for electrochemical tests. For the full wetting of the glass fiber separator, the normal electrolyte dosage in one coin cell is 100 μL. Electrochemical measurements were performed on a NEWARE battery testing system and Solartron multi-channel electrochemical workstation with the voltage range of 0.5-3.0 V (vs. $K^+/K$) in a thermostat at room temperature (25 °C). Cycling performance was conducted at specific currents of 167.5, 837.5, and 1675 mA g$^{-1}$, and rate performance was conducted at a series of specific currents in the order of 167.5, 335, 502.5, 837.5, 1675, 3350, 5025, and 167.5 mA g$^{-1}$, respectively.

## Symmetric cell measurements

The $K_2S_6$ electrolyte was fabricated by adding $K_2S$ and sulfur (molar ratio corresponds to the nominal stoichiometry of $K_2S_6$) into the tetra ethylene glycol dimethyl ether (TEGDME) with 3 M potassiumbis(tri-fluor-omethylsulfonyl)imide (KTFSI), and then stirring at 60 °C for 24 h. 100 μL obtained $K_2S_6$-contained electrolyte (0.052 M) with the identical anodes and cathodes of $W_{SA}$-$W_2C@NC$, $W_2C@NC$, and NC were assembled into the symmetric cells for the polysulfides conversion mechanism study.

## Visual cell measurements

The cathode materials ($W_{SA}$-$W_2C@NC/S$, $W_2C@NC/S$, or NC/S), MWCNTs, PVDF with a mass ratio of 6:3:1 was dispersed in NMP, and coated on carbon paper, followed by drying at 60 °C for overnight. The potassium foil that was stamped onto the stainless-steel collector was used as anode. Both anode and cathode were clamped by alligator clips, and the electrolyte is 4 M KTFSI in DME. Finally, the reaction vessels were sealed for galvanostatic charge-discharge measurements at 167.5 mA g$^{-1}$ on Solartron multi-channel electrochemical workstation.

## Computational methods

Details of the computational methods is provided within the Supplementary Information.

## Data availability

The data generated in this study are provided in the Supplementary Information and are available from the authors upon request. Source data are provided with this paper.

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

## Acknowledgements

We acknowledge financial supports from the National Natural Science Foundation of China (No. 52072257 (J. D.)), the National Key Research and Development Program of China (No. 2019YFE0118800 (Y. X.)), and the National Natural Science Foundation of China (No. 52372218 (J. D.), No. U22A20119 (W. H.)).

## Author contributions

W.H. and J.D. conceived the project and designed the experiments. W.S., X.Y., Z.H., and T.Z. carried out the synthesis and characterization of materials and the electrochemical measurements. W.S., Y.X. and J.S. performed the in-situ XRD measurements. W.S., H.W. and J.D. performed the DFT calculation. W.S., J.D., and W.H. wrote the paper. All authors discussed the results and commented on the paper.

## Competing interests

The authors declare no competing interests.
