## [Peer Review File · Nature Communications]

Synergistically optimizing potassium polysulfides for high performance potassium-sulfur batteriesREVIEWER COMMENTS

Reviewer #1 (Remarks to the Author):

Catalytic materials for metal-sulfur batteries are a booming area of research and the present study discusses the catalytic conversion of potassium polysulfides employing a WSA-W2C composite catalyst. Tungsten carbide has been reported as an efficient electrocatalyst in lithium-sulfur batteries previously with good catalytic conversion. Though the cycle life achieved in the study is on par with the current literature on potassium sulfur batteries, many critical aspects need to be revisited (please see the comments) in terms of design of experiments, reaction mechanism, and analysis. Given these, I do not recommend the publication of this work at this stage. The following comments could be helpful to improve the manuscript.

1. The introduction section of the manuscript should be revised to provide a better context for the study by discussing previous works in lithium-sulfur batteries where tungsten carbide or single atoms have been demonstrated as an electrocatalyst to enhance the performance of metal-sulfur batteries (Y. Wu et al. *Nano Energy*, 59, 636-643, 2019; B. Wei et al. *Chemistry-A European Journal*, 26, 68, 16057-16065, 2020; N. Shi et al. *Advanced Materials Interfaces*, 6, 9, 1802088, 2019). This will make the study more relevant to the field and help readers understand the motivation behind the current research.

2. The utilization of catalytic materials for enhancing polysulfide absorption and transformation is an efficient way to promote sulfur utilization in metal-sulfur batteries. However, the current study reports the preliminary results on the utilization of WSA-W2C as a catalyst in K-S batteries. The data set is preliminary and doesn't discuss the significance of the catalytic material (WSA-W2C) content in the obtained electrochemical performances.

(a) How critical is the amount of WSA and W2C to the sulfur content in enhancing the catalytic conversion of potassium sulfides?

(b) How reproducible is the study, with regard to tuning the formation of WSA and W2C in the WSA-W2C@NC/S composite?

(c) It is important to systematically investigate the sulfur utilization and cyclability in K-S batteries with varying catalyst (WSA-W2C) content. However, the authors should systematically investigate the impact of varying amounts of WSA-W2C catalyst on the sulfur content in the composite and the electrode to determine the optimal amount of catalyst required to enhance the catalytic conversion of potassium sulfides. This will help to improve the feasibility of using catalysts to promote sulfur utilization in metal-sulfur batteries.

3. The authors have compared WSA-W2C@NC/S, W2C@NC/S, and NC/S. The rationale behind comparing the WSA-W2C@NC/S sample with W2C@NC/S is not clear. Do these samples have similar W2C contents? The sulfur content in the WSA-W2C@NC/S, W2C@NC/S, and NC/S also vary significantly and are not comparable.

4. The formation of tungsten carbide W₂C catalyst should be confirmed with XRD. Please add the XRD of the WSA-W₂C@NC, W₂C@NC, WSA-W₂C@NC/S and W₂C@NC/S.

5. The authors have mentioned that the WSA-W₂C@NC/S composite achieved a sulfur utilization of 89% at 0.1 C. However, the authors should provide more detailed information on the electrochemical characterization of the electrodes by mentioning both the electrode weight (mg/cm²) and the total sulfur content in the electrodes (%). Furthermore, the sulfur content in the electrodes in the current study should be compared with previous works listed in Supplementary Table 2 to help readers better understand the level of achievement in the current study compared to previous literature.

6. The performance of the WSA-W₂C@NC/S composite obtained in the current study is similar to that of SPAN cathodes as the authors have mentioned. The limitation of the SPAN cathodes is mainly about the low sulfur content in the composite level which translates to the electrode level. The sulfur content in the WSA-W₂C@NC/S composite (TGA curves, SI) is lower than normal SPAN composites (typically 45-50%) reported. Have the authors tried to improve the sulfur content in the current composite?

7. The authors have mentioned that the first cathodic scan of WSA-W₂C@NC/S, has a strong peak at 0.836 V corresponding to the SEI formation and the sulfur activation. Please add relevant references that confirm this statement. Also, why do the initial CV curves in Fig. S14 lack the peaks at R1 and R2? The reaction mechanism should be discussed correlating to this observation.

8. Additional discussions on the sulfur content in the WSA-W₂C@NC/S, W₂C@NC/S, and NC/S composites based on the TGA curves should be added in the discussion part.

9. Discussions on the type of nitrogen doping and its impact on the polysulfide conversion and electrochemical performances observed in the samples could be added.

10. UV-visible spectroscopy is a simple and powerful technique to analyze the polysulfide adsorption and conversion capability of catalytic materials. Please add them to confirm the adsorption and conversion capability of WSA-W₂C@NC and compare them with that of W₂C@NC and NC.

Reviewer #2 (Remarks to the Author):

This is a super-impressive and -creative research.

Potassium-based energy storage technologies are the cutting-edge field and solution for the large-scale low-cost stationary electrical energy storage systems (ESSs), and the potassium – sulfur (K-S) battery is the extremely significant focus. However, K-S batteries still suffer from many key challenges, including the very low utilization of sulfur, the sluggish conversion kinetics of potassium polysulfides (KPSs) and

the resulting shuttling, as well as the difficult decomposition of solid discharge products.

In this work, authors proposed an innovative strategy of coupling polysulfides migration and conversion enhancements for improving the potassium sulfur electrochemistry in battery. The role of polysulfides transportation in the K-S cell is for the first time emphasized and optimized. The concept of achieving dual-functionality by screening and constructing WSA-W2C composite active site on sulfur host is enlightening. The performance of K⁺ storage by sulfur cathode also presents new information for the community. Moreover, the thermodynamic and kinetic mechanisms of potassium sulfur reaction influenced by the host functionalities were clearly demonstrated based on comprehensive experiment and calculation results.

Overall, the story falls in the scope of Nature communications and can bring new and inspiring ideas for the readers. Therefore, I strongly recommend it being accepted after a minor revision.

Comments:

1. It is impressive to see the cyclability of the cells at such high sulfur utilization level. The authors are better to further investigate the resistance changes in the K-S cells for explaining the sulfur cathode durability enhancement.
2. I notice that normally one transition state between initial and final states was determined for calculating the energy barriers (figure 1b, 4e, 5f). For more accurate calculation, two or more transition states should be considered in order to obtain accurate energy profiles.
3. The design of WSA-W2C composite is interesting. The EXAFS data in figure 1d proved the existence of single W atoms, but it will be helpful for deep structure-property correlation analysis if the authors can further determine the accurate configuration.
4. As supplemental analysis for the polysulfide conversion and the possible shuttling, the characteristics of the cycled potassium anode in different cells should be further revealed.
5. The authors are suggested to consider more control samples to verify the unique effect of WSA-W2C composite, for instance the single W atom based host containing no W2C.

Reviewer #3 (Remarks to the Author):

The work reported by W. Song et al. used WSA-W2C-NC composite as a sulfur for improving the performance of KSBs. The theoretical approach has been adopted to synthesize the composite for better electrochemical features. Further, a detailed analysis has been carried out to investigate the role of WSA-

W2C in polysulfide capture and conversion. After careful evaluation, the manuscript can be improved further with a major revision before acceptance for the publication. Specific comments are provided below:

1. Introduction should be modified to illustrate why WSA, W2C, and N-doped carbon have been adopted for this work.
2. For sulfur composite preparation: (a) Why 1:1 mass ratio is used? (b) Why has three-step heating been followed?
3. For electrode preparation: (a) Why was composite only 60% while 40% (CNT and binder) has been used, as this decreases the active material on the electrode? Currently, sulfur content on the electrode is only $0.6 \times 0.4 = 0.24$ fraction (=24%). (b) What is the sulfur loading of the electrode?
4. Infusion of sulfur is known to increase the interlayer spacing. In this case, why does it not affect the interlayer spacing of W2S?
5. The sulfur composites have been synthesized at the melting point of sulfur. How do the composites show the amorphous nature of sulfur?
6. The electrochemical performance should be performed with higher sulfur loading ($> 2.5 \text{ mg cm}^{-2}$).
7. The author should also investigate the interaction of long-chain polysulfide with the host material using polysulfide adsorption and symmetric cell studies (refer to this: Carbon 212 (2023) 118173, ACS Appl. Energy Mater. 2023, 6, 5, 3042–3051, Electrochimica Acta 422 (2022) 140531).

Response Letter for Manuscript “Synergy of facilitated migration and catalytic conversion towards potassium polysulfides enables high performance potassium-sulfur batteries (NCOMMS-23-28977)”

We are grateful to the referees for their highly constructive and detailed reviews. The manuscript has been revised in accordance with all their comments. These positive changes and additions are detailed in the Response Letter and are highlighted in blue font in the revised main text, figures (legends) and Supporting Information.

Reviewer #1:

General comments:

Catalytic materials for metal-sulfur batteries are a booming area of research and the present study discusses the catalytic conversion of potassium polysulfides employing a $W_{SA}-W_2C$ composite catalyst. Tungsten carbide has been reported as an efficient electrocatalyst in lithium-sulfur batteries previously with good catalytic conversion. Though the cycle life achieved in the study is on par with the current literature on potassium sulfur batteries, many critical aspects need to be revisited (please see the comments) in terms of design of experiments, reaction mechanism, and analysis. Given these, I do not recommend the publication of this work at this stage. The following comments could be helpful to improve the manuscript.

Author response:

We sincerely appreciate the reviewer’s careful reading of our work and their insightful input. Accordingly, we have revised the manuscript to address all the points that were raised. The revised sections are written in blue font, both in the response letter and in the revised manuscript, text and figures. We feel that the revised work is much improved as a result. Please refer to the detailed point by point responses below. We would like to again thank you for considering our enterprise.

Referee comment #1:

The introduction section of the manuscript should be revised to provide a better context for the study by discussing previous works in lithium-sulfur batteries where tungsten carbide or single atoms have been demonstrated as an electrocatalyst to enhance the performance of metal-sulfur batteries (Y. Wu et al. Nano Energy, 59, 636-

643, 2019; B. Wei et al. *Chemistry-A European Journal*, 26, 68, 16057-16065, 2020; N. Shi et al. *Advanced Materials Interfaces*, 6, 9, 1802088, 2019). This will make the study more relevant to the field and help readers understand the motivation behind the current research.

Author response #1:

We sincerely thank the referee for this excellent suggestion. We fully agree that the previous studies of tungsten based electrocatalysts for lithium-sulfur batteries and sodium-sulfur batteries are highly instructive and should be discussed in the Introduction. Tungsten of various forms (single atom, cluster, nanoparticle, compounds) were reported highly catalytically active for the conversion of both lithium and sodium polysulfides. To highlight the previous progresses, we revised the Introduction discussing the related studies of tungsten based sulfur cathodes in metal-sulfur batteries. The inspiring studies mentioned by the referee were included in the revised Introduction. The revisions are as below in blue font.

Introduction

“.....

Some pioneering efforts were made to innovate functional sulfur hosts for KSBs. Most of the hosts are carbon-based, taking advantage of the good electrical conductivity, diverse surface chemistry, and versatile porosity of carbon.^{6, 7} The sulfur confinement is the primary functionality of the host. Physically confining sulfur in the carbon porosity is a popular strategy.^{8, 9} For instance, microporous carbons confining small molecular (S₁₋₃) exhibited promising capacity and cyclability in KSBs. Nonetheless, the limitation of pore volume normally sacrificed the sulfur loading.^{7, 10} The carbon hosts employed were routinely functionalized by nitrogen moieties, which are effective of enhancing the sulfur confinement via covalent bonding at heterointerfaces.¹¹ In addition, the chemical confinement strategy was also widely reported.¹²⁻¹⁴ Representatively, sulfurized polyacrylonitrile (SPAN) utilizing the covalent sulfur-carbon bonds greatly inhibited KPSs shuttling.¹⁴⁻¹⁷ The chemical confinement also includes the chemical adsorption between KPSs and certain polar species of the host, such as heteroatom groups, nanostructured compounds, atomic metal assemblies.^{7, 18, 19} For instance, single metal atoms^{18, 20, 21} and metal carbides²²⁻²⁵ were reported providing adsorption capability towards polysulfides.

The functionality of catalytic conversion towards KPSs for sulfur host attracted increasing attention in recent years. Some highly enlightening studies introduced catalytic sites in the hosts in order to improve the kinetics of KPSs conversion, especially the solid-state reactions like K_2S oxidation.²⁶ Metallic single atoms^{18, 26} and metallic atom clusters²⁷ were reported as electrocatalytic species in the hosts, which suppressed the long-chain KPSs shuttling and reduced the polarization of solid-state sulfide conversion, thereby enhancing the sulfur utilization, rate and cycling capability. The design of catalytic materials for KPSs can be inspired by the prior achievements in Li/Na-sulfur batteries. The metal element of various forms (*e.g.* single atom, atomic cluster, compounds) that can deliver general catalytic capability towards both lithium and sodium polysulfides deserves particular attention. For example, tungsten can be catalytically active in Li-S batteries (tungsten single atom²⁰, tungsten carbide²³⁻²⁵, tungsten sulfide^{28, 29}) and Na-S batteries (tungsten nanoparticle³⁰). The successful applications of tungsten-based catalysts in metal sulfur batteries were largely ascribed to the special electronic structure of tungsten atomic assemblies and compounds.³¹ These insights are highly instructive for current KSBs explorations.

.....”

References

11. *Adv. Energy Mater.* **10**, 2000931 (2020).
18. *Angew. Chem. Int. Ed.* **62**, e202301681 (2023).
20. *Angew. Chem. Int. Ed.* **60**, 15563 (2021).
21. *Adv. Mater.* **35**, 2208873 (2022).
22. *Adv. Mater.* **34**, e2106572 (2022).
23. *Nano Energy* **59**, 636 (2019).
24. *Chem. Eur. J.* **26**, 16057 (2020).
25. *Adv. Mater. Interfaces* **6**, 1802088 (2019).
28. *Adv. Energy Mater.* **10**, 2000091 (2020).
29. *Adv. Energy Mater.* **7**, 1601843 (2017).
30. *Adv. Sci.* **9**, 2105544 (2022).

Referee comment #2:

The utilization of catalytic materials for enhancing polysulfide absorption and transformation is an efficient way to promote sulfur utilization in metal-sulfur batteries. However, the current study reports the preliminary results on the utilization of $W_{SA}-W_2C$ as a catalyst in K-S batteries. The data set is preliminary and doesn't

discuss the significance of the catalytic material ($W_{SA}-W_2C$) content in the obtained electrochemical performances.

(a) How critical is the amount of W_{SA} and W_2C to the sulfur content in enhancing the catalytic conversion of potassium sulfides?

(b) How reproducible is the study, with regard to tuning the formation of W_{SA} and W_2C in the $W_{SA}-W_2C@NC/S$ composite?

(c) It is important to systematically investigate the sulfur utilization and cyclability in K-S batteries with varying catalyst ($W_{SA}-W_2C$) content. However, the authors should systematically investigate the impact of varying amounts of $W_{SA}-W_2C$ catalyst on the sulfur content in the composite and the electrode to determine the optimal amount of catalyst required to enhance the catalytic conversion of potassium sulfides. This will help to improve the feasibility of using catalysts to promote sulfur utilization in metal-sulfur batteries.

Author response #2:

We sincerely appreciate the referee for carefully reading the manuscript and providing highly valuable guidance. The referee raised a very important point that the content of the $W_{SA}-W_2C$ species for affecting the electrochemical performance of K-S batteries. We fully agree with the referee for this point. In fact, we have considered this point at the starting stage of our study. We have conducted control experiments to systemically investigate the role of $W_{SA}-W_2C$ content in affecting K-S electrochemistry performance. The $W_{SA}-W_2C@NC/S$ specimen reported in the manuscript is the optimized candidate with the optimal amount of $W_{SA}-W_2C$ specie. We didn't include the data of other control specimens in the original manuscript due to the length limitation of the manuscript. In order to address this comment, we organized the data of other control specimens and demonstrate the role of $W_{SA}-W_2C$ content in affecting K-S electrochemistry performance in the revised manuscript.

(a) The amount of $W_{SA}-W_2C$ is indeed a critical factor. To investigate the amount of $W_{SA}-W_2C$ in affecting the catalytic conversion of potassium polysulfides (KPSs), we prepared $W_{SA}-W_2C-L@NC$ having lower $W_{SA}-W_2C$ content than that of $W_{SA}-W_2C@NC$, and $W_{SA}-W_2C-H@NC$ having higher $W_{SA}-W_2C$ content than that of $W_{SA}-W_2C@NC$. The host-sulfur composites ($W_{SA}-W_2C-L@NC/S$, $W_{SA}-W_2C-H@NC/S$) were also prepared following identical sulfur impregnation procedure as $W_{SA}-W_2C@NC/S$. These composites with different amount of $W_{SA}-W_2C$ species were

systematically analyzed and compared in terms of morphology, tungsten specie structures, sulfur content and electrochemical performances (sulfur utilization and cyclability).

New Supplementary Figure 6. **a** SEM image, **b** HRTEM image, **c** HAADF-STEM image and **d** EDS mapping of $W_{SA}-W_2C-L@NC$. **e** SEM image, **f** HRTEM image, **g** HAADF-STEM image and **h** EDS mapping of $W_{SA}-W_2C-H@NC$.

The morphologies of $W_{SA}-W_2C-L@NC$ and $W_{SA}-W_2C-H@NC$ were similar to that of $W_{SA}-W_2C@NC$, which are basically polyhedron particles around 200 nm in size (Supplementary Fig. 6a and 6e). High resolution TEM images of Supplementary Fig. 6b and 6f revealed the existence of W_2C nanocrystals. Overall, $W_{SA}-W_2C-L@NC$ has lower density of W_2C nanocrystals than that of $W_{SA}-W_2C-H@NC$. As shown in Supplementary Fig. 6c and 6g, the high resolution HAADF images obtained in ac-STEM prove the presence of W single atoms (highlighted by solid circles) in both host materials. The corresponding EDS mapping also proved this point (Supplementary Fig. 6d and 6h).

New Supplementary Figure 11. **a** SEM image, **b** HRTEM image, and **c** EDS mapping of $W_{SA}-W_2C-L@NC/S$. **d** SEM image, **e** HRTEM image, and **f** EDS mapping of $W_{SA}-W_2C-H@NC/S$. **g** TGA curves of $W_{SA}-W_2C-L@NC/S$ and $W_{SA}-W_2C-H@NC/S$. **h** Nitrogen adsorption-desorption isotherms and **i** XRD patterns of $W_{SA}-W_2C-L@NC$, $W_{SA}-W_2C-H@NC$, $W_{SA}-W_2C-L@NC/S$, and $W_{SA}-W_2C-H@NC/S$.

According to the SEM images of $W_{SA}-W_2C-L@NC/S$ and $W_{SA}-W_2C-H@NC/S$ (Supplementary Fig. 11a and 11d), sulfur have fully impregnated into the hosts. The size and crystal structures of the tungsten carbides remain unchanged (Supplementary Fig. 11b and 11e). $W_{SA}-W_2C-L@NC/S$ and $W_{SA}-W_2C-H@NC/S$ have similar sulfur content of 42.12% and 40.75% determined by TGA (Supplementary Fig. 11g). According to nitrogen adsorption-desorption data, the $W_{SA}-W_2C$ content cannot markedly affect the porous structure of the carbons (Supplementary Fig. 11h). The sulfur in amorphous texture was accommodated inside the porosity of the hosts (Supplementary Fig. 11i).

New Supplementary Figure 24. Electrochemical performance of KSBs employing $W_{SA}-W_2C-L@NC/S$ and $W_{SA}-W_2C-H@NC/S$ cathodes. **a** GCD curves at 0.1C. **b** Cyclability at 0.5C and 1C.

Supplementary Figure 24a displayed the galvanostatic charge/discharge (GCD) profiles of $W_{SA}-W_2C-L@NC/S$ and $W_{SA}-W_2C-H@NC/S$ cathodes. The sulfur utilizations of $W_{SA}-W_2C-L@NC/S$ and $W_{SA}-W_2C-H@NC/S$ are respectively 84.5% and 76.1%, which are lower than that of $W_{SA}-W_2C@NC/S$. Supplementary Figure 24b shows the cycling performances at 0.5 and 1C. Comparing to $W_{SA}-W_2C@NC/S$, $W_{SA}-W_2C-L@NC/S$ and $W_{SA}-W_2C-H@NC/S$ cathodes delivered inferior cyclability. According to the electrochemical performance, $W_{SA}-W_2C@NC/S$ has the optimal content of $W_{SA}-W_2C$, and therefore was employed for further K-S electrochemistry mechanism studies.

(b) Our experiments have excellent reproducibility. The tuning of W_{SA} and W_2C formation in $W_{SA}-W_2C@NC$ during the synthesis is highly controllable. Different batches of specimens synthesized by identical synthesis procedure demonstrated good batch consistency. With a wide range of tungsten precursor dosage, well-defined composites of W_{SA} and W_2C nanocrystals can be obtained, as proved by the synthesis processes and the characterization of $W_{SA}-W_2C-H/-L@NC$ host materials.

(c) We fully agree with the referee on this point. We have systematically investigated the impact of varying amounts of $W_{SA}-W_2C$ species on the K-S electrochemical performance and finally determined the optimal amount. The data and discussions were present above. Please refer to the responses to *Comment #2 (a)* above.

The relevant texts revisions are as below in blue font.

“.....

Single tungsten atoms distribute over the whole NC substrate and embrace W₂C nanocrystals with higher density.....For determining the optimal content of W_{SA}-W₂C in the hosts, specimens with higher W_{SA}-W₂C content (W_{SA}-W₂C-H@NC) and lower W_{SA}-W₂C content (W_{SA}-W₂C-L@NC) were also prepared. The W_{SA} and W₂C species in these two control specimens could be well identified by SEM, HRTEM and HADDF-STEM images (Supplementary Fig. 6).

.....”

“.....

According to HRTEM images, the size and crystal structures of the tungsten carbides remain unchanged after sulfur loading for both W_{SA}-W₂C@NC/S (Fig. 2e) and W₂C@NC/S (Supplementary Fig. 10). Control specimens of W_{SA}-W₂C-H@NC/S and W_{SA}-W₂C-L@NC/S display the same phenomenon, as revealed in Supplementary Fig. 11. Moreover, X-ray photoelectron spectroscopy (XPS) revealed the interaction between sulfur and the host.^{28, 42, 43} In Fig. 2f of W 4f spectra, the sulfur loading induced an elevated proportion of W⁵⁺ and an increase in average valence from +3.43 to +4.07 for W_{SA}-W₂C@NC/S. By contrast, the W 4f in W₂C@NC/S hardly changed (Supplementary Fig. 12). This phenomenon indicates that the interaction between sulfur and atomic-level homogeneously dispersed W_{SA} is more pronounced, which also suggests the sulfur distribution all over the entire host.

.....”

“.....

The extremely low capacities of the pure hosts (Supplementary Fig. 21) verified that the capacities of cathodes are derived from the reversible sulfur redox. Control cathodes of W_{SA}-W₂C-H@NC/S and W_{SA}-W₂C-L@NC/S delivered capacities of 1274.4 and 1416.1 mAh g⁻¹ (Supplementary Fig. 24), suggesting W_{SA}-W₂C@NC/S has the optimal content of W_{SA}-W₂C.

.....”

“.....

Afterwards, the cathodes kept stable cycling for 200 and 97 cycles, resulting in capacity retention ratio of 49.4 and 56.8%. W_{SA}-W₂C-H@NC/S and W_{SA}-W₂C-L@NC/S showed inferior cyclability, again proving the optimal content of W_{SA}-W₂C in W_{SA}-W₂C@NC (Supplementary Fig. 24).

.....”

Supporting information.

“**Synthesis of $W_{SA}-W_2C-H@NC$ and $W_{SA}-W_2C-L@NC$.** The synthesis of $W_{SA}-W_2C-H@NC$ and $W_{SA}-W_2C-L@NC$ followed the same procedure of $W_{SA}-W_2C@NC$ except different amounts of $Na_2WO_4 \cdot 2H_2O$ (0.6 g for $W_{SA}-W_2C-H@NC$ and 0.1 g for $W_{SA}-W_2C-L@NC$) were added for preparing WO_4^{2-} -ZIF-8 precursors.”

Referee comment #3:

The authors have compared $W_{SA}-W_2C@NC/S$, $W_2C@NC/S$, and NC/S . The rationale behind comparing the $W_{SA}-W_2C@NC/S$ sample with $W_2C@NC/S$ is not clear. Do these samples have similar W_2C contents? The sulfur content in the $W_{SA}-W_2C@NC/S$, $W_2C@NC/S$, and NC/S also vary significantly and are not comparable.

Author response #3:

We sincerely thank the referee for the highly instructive comments. The referee mentioned a very important point of making comparison between $W_{SA}-W_2C@NC/S$ and $W_2C@NC/S$ cathodes. The reason for making this control experiment is based on our key strategy of designing the tungsten functional species on host materials.

(1) Based on our previous research experiences, we concluded that incorporating simply the catalytic species on host materials is insufficient for achieving high performance K-S electrochemistry given the inevitable mismatch in the distribution between sulfur and catalytic sites. Therefore, we propose that there should be extra active species which can accelerate the KPSs migration over host materials. Guided by this thinking, tungsten single atom (W_{SA}) and W_2C were screened out from various tungsten species for the functionality of KPSs migration facilitating and conversion catalyzing, respectively. The $W_{SA}-W_2C@NC$ host possessing both W_{SA} and W_2C is expected to deliver KPSs migration and conversion bi-functionality. Whereas the W_{SA} free $W_2C@NC$ host has only the W_2C catalytic sites. Following this rationale, therefore, the comparison between $W_{SA}-W_2C@NC/S$ sample and $W_2C@NC/S$ sample is to demonstrate the critical role of synergizing the KPSs migration and conversion functionality for K-S electrochemical performance enhancement. This rationale was discussed systemically in the manuscript.

(2) The referee mentioned the W_2C contents in $W_{SA}-W_2C@NC$ and $W_2C@NC$. To answer this question, we proposed a reasonable method to quantify the W_2C

content based on TEM images. Figure R1a,b,c show three randomly selected regions of $W_{SA}-W_2C@NC$, and Figure R1d,e,f show three randomly selected regions of $W_2C@NC$. Based on these images, the populations and diameters of W_2C nanocrystals were counted. Because the regions for counting are randomly selected from the specimen under TEM, the numbers of W_2C obtained from these regions have reasonable statistical significance to reflect the entire specimen. Figure R1 includes the quantity and diameters of W_2C nanocrystals, based on which the total sizes of W_2C in each region can be calculated. Basically, the integrated W_2C sizes for $W_{SA}-W_2C@NC/S$ and $W_2C@NC/S$ are quite similar (Figure R2).

Figure R1. Statistics of W_2C nanoparticles on **a-c** $W_{SA}-W_2C@NC$ and **d-f** $W_2C@NC$.

Figure R2. Integral areas of W₂C nanoparticles in the regions shown in Figure R1.

(3) The referee also mentioned that the difference in sulfur content in three composites making them not comparable. The sulfur contents are 43.93%, 40.56% and 37.91% determined by TGA. We adopt a criterion to evaluate the absolute sulfur content difference by calculating the largest difference between medium content and largest/lowest contents in control specimens. Based on this criterion, the absolute difference is 3.37% in our case. We respectfully claim that this sulfur content difference among specimens is fairly low in control experiments for Na-S and K-S battery studies. Supplementary Table 2 displays the sulfur contents of control specimens for plenty of previous high-impact literatures of Na/K-S batteries. Among these data, the absolute difference of 3.37% in our case is at the lowest level.

New Supplementary Table 2. The analysis of sulfur contents in control samples in previous Na/K-S literatures.

Battery system	Control samples	Sulfur content	Difference	Measurement	Reference
K-S batteries	S@Co-NC S@NC S@SA-NC	49.3% 51.2% 56.8%	5.6%	TGA	J. Am. Chem. Soc. 143, 16902-16907 (2021)
	S@Cu-N ₄ S@NC	56.7 % 50.7%	6%	TGA	Angew. Chem. Int. Ed. 62, e202301681 (2023)
Na-S batteries	S@HC S@CoS ₂ /NC S@Co ₁ -CoS ₂ /NC	34% 50% 54%	16%	TGA	Angew. Chem. Int. Ed. 61, e202200384 (2022)
	Mn ₁ @NC@S Fe ₁ @NC@S Ni ₁ @NC@S Ge ₁ @NC@S Pt ₁ @NC@S Ru ₁ @NC@S	40% 68% 46% 17% 54% 53%	29%	TGA	Angew. Chem. Int. Ed. 132, 22355-22362 (2020)
	S@FeNi ₃ @HC S@Ni@HC S@HC	42% 42.5% 50.6%	8.1%	TGA	ACS Nano 15, 15218-15228 (2021)
	core-shell ZnS@S ZCS@S	47% 57%	10%	TGA	ACS Nano 14, 7259-7268 (2020)

CoS ₂ @S	58%				
S/TiN-TiO ₂ @MCCFs	56.9%	5.4%	TGA	ACS Nano 15, 5639-5648 (2021)	
S/MCCFs	51.5%				
MCPS1	47%	16%	TGA	Nat. Commun. 7, 11722 (2016)	
MCPS2	63%				
S/CoHC	48%	17%	TGA	Nat. Commun. 9, 4082 (2018)	
S@Co _n -HC	47%				
S@HC	30%				
NPCTs/S	47%	9%	TGA	Nat. Commun. 10, 4793 (2019)	
NiS ₂ @NPCTs/S	56%				
S@NCFs	45%	9%	TGA	Adv. Sci. 7, 1902617 (2020)	
S@Ni-NCFs	36%				
MMPCS-700@S	40.3%	3.7%	TGA	Adv. Mater. 34, 2108363 (2022)	
MMPCS-800@S	43.8%				
MMPCS-900@S	36.0%				
CN/Au/S	56.5%	3.6%	TGA	Energy Environ. Sci. 13, 562-570 (2020)	
CN/S	52.9%				
Y SAs/NC-S	67.4%	3.3%	TGA	J. Am. Chem. Soc. 144, 18995-19007 (2022)	
NC-S	64.1%				
155S	72%	28%	TGA	Nano-Micro Lett. 13, 121 (2021)	
300S	44%				

In order to address this concern of the referee, we prepared new specimens named W₂C@NC/S-New and NC/S-New, which have almost identical sulfur contents to that of W_{SA}-W₂C@NC/S. As shown in Supplementary Fig. 20a, the sulfur contents of W₂C@NC/S-new, NC/S-new and W_{SA}-W₂C@NC/S are very close (*i.e.* 40.21%, 41.01%, 40.56%). Supplementary Figure 20b,c displayed the galvanostatic charge/discharge (GCD) profiles of the three specimens. The profiles of W₂C@NC/S-New and NC/S-New are highly overlapped with those of W₂C@NC/S and NC/S, suggesting that the slight difference in sulfur content has not affected the K-S electrochemical performance. The sulfur content difference among three specimens doesn't change the conclusion in our study.

New Supplementary Figure 20. **a** TGA data of NC/S-New, W₂C@NC/S-New and W_{SA}-W₂C@NC/S. **b** GCD profiles of NC/S and NC/S-New. **c** GCD profiles of W₂C@NC/S and W₂C@NC/S-New.

The relevant revision in the revised manuscript is as follow in blue font.

“.....

The discrepancy in sulfur content among three cathodes is below 3.4%, which is at the lowest level in the field (Supplementary Table 2). The cathodes with such sulfur content discrepancy deliver negligible difference in electrochemistry (Supplementary Fig. 20).

.....”

Referee comment #4:

The formation of tungsten carbide W_2C catalyst should be confirmed with XRD. Please add the XRD of the $W_{SA}-W_2C@NC$, $W_2C@NC$, $W_{SA}-W_2C@NC/S$ and $W_2C@NC/S$.

Author response #4:

We fully agree with the referee that XRD characterization should be conducted for the specimens. We also respectfully state that XRD may be limited to detect the very small W_2C nanocrystals. To address this comment, we added the XRD patterns in the revised manuscript. According to Supplementary Fig. 5, the XRD patterns display only the amorphous textures of carbon matrix in the specimens. The small and relatively low loading W_2C nanocrystals cannot be detected by XRD. Therefore, we employed TEM based techniques including selected area diffraction pattern (SAED), high resolution TEM and ac-TEM images to confirm the presence of W_2C nanocrystals.

The corresponding revisions are in blue font.

New Supplementary Figure 5. XRD patterns of **a** $W_{SA}-W_2C@NC$ and $W_2C@NC$, **b** $W_{SA}-W_2C@NC/S$ and $W_2C@NC/S$.

“.....

Single tungsten atoms distribute over the whole NC substrate and embrace W₂C nanocrystals with higher density. Of note, the small-size and relatively low loading W₂C nanocrystals cannot be detected by XRD (Supplementary Fig. 5).

.....”

“.....

In HRTEM and HAADF-STEM images (Supplementary Figs. 4b and 7), only tungsten carbide (W₂C) nanocrystals were observed with scarcely any single tungsten atom present. Similar to W_{SA}-W₂C@NC, the W₂C nanocrystals are too small to be detected by XRD (Supplementary Fig. 5).

.....”

Referee comment #5:

The authors have mentioned that the W_{SA}-W₂C@NC/S composite achieved a sulfur utilization of 89% at 0.1 C. However, the authors should provide more detailed information on the electrochemical characterization of the electrodes by mentioning both the electrode weight (mg/cm²) and the total sulfur content in the electrodes (%). Furthermore, the sulfur content in the electrodes in the current study should be compared with previous works listed in Supplementary Table 2 to help readers better understand the level of achievement in the current study compared to previous literature.

Author response #5:

We fully agree with the referee that the sulfur content (%) and active material loading (mg cm⁻²) in the electrode should be provided as discussing the electrochemical performance of sulfur cathode in K-S battery. Following the referee's suggestion, we made a comprehensive table (Supplementary Table 3) to compare W_{SA}-W₂C@NC/S with the state-of-the-art sulfur cathodes in K-S batteries in terms of electrochemistry parameters and sulfur loading parameters, in order to highlight the advances of the results in this manuscript.

First, as shown in Supplementary Table 3, the sulfur content in W_{SA}-W₂C@NC/S composite is 40.56%, which is higher than the previous reported microporous

carbon/sulfur composite cathodes (Ref. 10, 11, 12) and on par with the previous SPAN-based cathodes (Ref. 13, 14, 15, 16, 17). Given this sulfur content, our study achieved the highest sulfur utilization of 89.8% by far. The capacities at higher rates, e.g. 1214 mAh g⁻¹ at 0.5C and 1059 mAh g⁻¹ at 1C are also outstanding comparing to the state-of-the-art K-S batteries.

New Supplementary Table 3. Comparison between W_{SA}-W₂C@NC/S and the state-of-the-art cathodes for KSBs.

Cathodes	Sulfur content (%)	Sulfur loading (mg cm ⁻²)	Maximum Sulfur utilization	Rate performance (Capacity (mAh g ⁻¹) @ Current (mA g ⁻¹))	Ref.
W _{SA} -W ₂ C@NC/S	40.56%	~1.0/ ~2.8	89.8%/ 88.2%	1504 @ 167.5 1214 @ 837.5 1059 @ 1675	This work
C/S composite	18.6%	0.5–1.0	77.2%	1293.3 @ 20 741.2 @ 2000	10
PCNF/S	25%	0.5–1.0	83.0%	1392 @ 20 1138 @ 200	11
S@P-NCF	37%	~0.8	87.7%	1470 @ 337 560 @ 3370	12
SPAN	39.52%	Not reported	31.7%	532 @ 35	13
SPAN	38%	~1.0	42.4%	710 @ 47.5 218 @ 285	14
SPAN	36%	~0.8	58.9%	987 @ 837.5	15
I-S@pPAN	~42%	~1.0	86.1%	1442 @ 167.5	16
FS-SPAN	33%	~0.8	80.2%	1345 @ 70 680 @ 1000	17
γS-CNFs	~50%	0.69	52.8%	885 @ 167.5 66 @ 3350	18
CCS	39.25%	0.39–0.59	26.9%	440 @ 150 94 @ 1000	19
EAMC12	37.8%	1.5–2.0	69.5%	1165 @ 335 572 @ 1675	20
S@SA-NC	51.2%	~1.0	75.6%	1266 @ 337 868 @ 837.5	21
CCS@CBC_450 (3 M KTF3SI in TEGDME)	40%	~1.5	42.4	711 @ 100 130 @ 1000	22
S/CNF (1 M KCF ₃ SO ₃ in TEGDME)	27.3%	~1.0	67.2%	1126 @ 167.5 938 @ 335 780 @ 558.3	23
CNT/S (3 M KFSI in DME)	25.5%	~0.2	18.6%	311 @ 50 94 @ 500	24

ACF-1500@S (3 M KFSI in DME)	15%	~1.3	18.1%	303 @ 50	25
S ₈ /VC (0.3 M Cu(TFSI) ₂ -0.1 M KTFSI in Me-Im)	Not reported	1–1.2	50.6%	847 @ 150	26
0.05 M K ₂ S ₅₋₆ catholyte (0.5 M KTFSI in DEGDME)	Not reported	Not reported	22.3%	374 @ 55.8 107 @ 1116	27
0.02 M K ₂ S ₅ catholyte (1 M KTFSI in DEGDME)	Not reported	~0.128	33.2%	556 @ 117	28

References

10. *ACS Nano* **13**, 2536 (2019).
11. *J. Mater. Chem. A* **8**, 10875 (2020).
12. *Nanomaterials* **12**, 3968 (2022).
13. *Mater. Lett.* **242**, 5 (2019).
14. *Chem. Commun.* **54**, 2288 (2018).
15. *J. Mater. Chem. A* **6**, 14587 (2018).
16. *Chem. Commun.* **55**, 5267 (2019).
17. *Small Methods* **6**, 2100899 (2022).
18. *J. Mater. Chem. A* **11**, 15924 (2023).
19. *Electrochim. Acta* **293**, 191 (2019).
20. *Chem. Commun.* **57**, 1490 (2021).
21. *J. Am. Chem. Soc.* **143**, 16902 (2021).
22. *ACS Sustain. Chem. Eng.* **10**, 16634 (2022).
23. *Energy Stor. Mater.* **15**, 368 (2018).
24. *J. Electron. Mater.* **50**, 3037 (2021).
25. *J. Power Sources* **480**, 228874 (2020).
26. *J. Mater. Chem. A* **7**, 20584 (2019).
27. *ACS Energy Lett.* **3**, 540 (2018).
28. *Angew. Chem. Int. Ed.* **61**, e202200606 (2022).

Second, for routine electrochemical measurement of our study, the typical sulfur loading in the electrodes is $\sim 1 \text{ mg cm}^{-2}$. According to Supplementary Table 3, the sulfur loadings of 1 mg cm^{-2} or lower were widely applied in the previous researches of Na-S and K-S batteries in the field, which is also a proper sulfur loading level for studying the Na-S/K-S electrochemistry. For W_{SA}-W₂C@NC/S based cathode, we can also increase the sulfur loading over 2.5 mg cm^{-2} , which has never been reported in K-S battery system to date. As shown in Supplementary Fig. 23, even in the high sulfur loading of 2.86 mg cm^{-2} , the sulfur utilization can still reach as high as 88.2%, which is still higher than the state-of-the-art results under low sulfur loading conditions.

New Supplementary Figure 23. a GCD profiles curves at 0.1C and **b** cyclability of KSBs employing $W_{SA}-W_2C@NC/S$ cathodes with high sulfur loadings.

Therefore, the aforementioned comparison highlights the advances of our $W_{SA}-W_2C@NC/S$ cathodes for K-S battery in the community. The relevant revision in the revised manuscript is as follow in blue font.

“.....

The extremely low capacities of the pure hosts (Supplementary Fig. 21) verified that the capacities of cathodes are derived from the reversible sulfur redox. $W_{SA}-W_2C@NC/S$ electrodes with high sulfur loading of 2.86 and 3.61 $mg\ cm^{-2}$ delivered slightly lower capacities of 1477.8 and 1221.9 $mAh\ g^{-1}$ (Supplementary Fig. 23), which are still higher than most previously reported cathodes with low sulfur loading (Supplementary Table 3).

.....”

“.....

The $W_{SA}-W_2C@NC/S$ cathode delivered a high discharge capacity of 1504 $mAh\ g^{-1}$ based on the thermogravimetric analysis (TGA) determined sulfur content (40.56%) (Supplementary Fig. 19). The absolute sulfur contents above 40% are higher than that for reported sulfur-microporous carbon composites and on par with SPANs for KSBs (Supplementary Table 3).

.....”

Supporting Information.

“.....

The electrolyte is 0.8 M KPF_6 in EC/DEC with a volume ratio of 1:1. The typical sulfur loading of the electrode is ca. 1 $mg\ cm^{-2}$. Electrodes with high sulfur loading of ca. 2.8 and ca. 3.6 $mg\ cm^{-2}$ were also prepared for electrochemical tests. For the full

wetting of the glass fiber separator, the normal electrolyte dosage in one coin cell is 100 μL .

.....”

Referee comment #6:

The performance of the $W_{SA}\text{-}W_2\text{C@NC/S}$ composite obtained in the current study is similar to that of SPAN cathodes as the authors have mentioned. The limitation of the SPAN cathodes is mainly about the low sulfur content in the composite level which translates to the electrode level. The sulfur content in the $W_{SA}\text{-}W_2\text{C@NC/S}$ composite (TGA curves, SI) is lower than normal SPAN composites (typically 45-50%) reported. Have the authors tried to improve the sulfur content in the current composite?

Author response #6:

We thank the referee for raising this important point. We fully agree that SPAN is an important candidate for metal-sulfur battery and should be considered as a criterion to judge the performance of other sulfur cathode materials. Therefore, we made a comprehensive comparison between $W_{SA}\text{-}W_2\text{C@NC/S}$ and the previously reported SPAN cathodes for K-S batteries in terms of sulfur content, sulfur utilization, specific capacity, cyclability and rate performance. All the SPAN cathodes reported for K-S batteries by far are included for the comparison. The detailed information is listed in Table R1. For the sulfur content comparison, we also included the literatures of SPAN cathodes in Na-S batteries to highlight the sulfur content and utilization. Other aspects of performances of Na-S batteries were not compared to $W_{SA}\text{-}W_2\text{C@NC/S}$ due to the system difference.

Table R1. Comprehensive comparison between $W_{SA}\text{-}W_2\text{C@NC/S}$ and previously reported SPAN based cathodes in literatures.

Battery system	Sample	Sulfur content (%)	Sulfur utilization (%) @ Current ($\text{mA g}_{\text{sulfur}}^{-1}$)	Cyclability	Rate performance @ Current ($\text{mA g}_{\text{sulfur}}^{-1}$)	Ref.
K-S batteries	$W_{SA}\text{-}W_2\text{C@NC/S}$	40.56%	89.8% @ 167.5	56.8% upon 98 cycles @ 1675	1504 @ 167.5 1214 @ 837.5 1059 @ 1675	This work
	SPAN composite	~38%	76.4% @ 13.3	50.7% upon 100 cycles @ 411	950.6 @ 329 82.7 @ 1974	1

	CCS	39.25%	26.9% @ 150	35% upon 200 cycles @ 150	440 @ 150 94 @ 1000	2
	SPAN	39.52%	41.8% @ 49.4	26.3% upon 100 cycles @ 35	532 @ 35	3
	FS-SPAN	33%	83.6% @ 70	44.6% upon 235 cycles @ 1000	1345 @ 70 680 @ 1000	4
	I-S@pPAN	42%	75.8% @ 167.5	61.4% upon 100 cycles @ 167.5	1383 @ 167.5 978 @ 837.5 788 @ 1675	5
	SPAN nanocomposite	45.5%	62.7% @ 370	37.6% upon 100 cycles @ 837.5	371 @ 837.5 173 @ 1675	6
Na-S batteries	SPAN/PDA	24.92%	83.6% @ 335	/	/	7
	SPAN	38.18%	/	/	/	8
	Se _{0.08} S _{0.92} @pPAN	36.88%	59.7% @ 100	/	/	9
	C-PANS	31.42%	71.9% @ 113.7	/	/	10
	SPAN	40.24%	68.0% @ 335	/	/	11
	CS-DPAN	~15%	67.5% @ 167.5	/	/	12
	SPAN web	37.4%	89.5% @ 267	/	/	13
	SPAN composite	42.4%	61.5% @ 335	/	/	14
	SPAN	44%	65.7% @ 100	/	/	15
	SPAN web	41%	47.3% @ 16.75	/	/	16
	H-SPAN film	41.19%	71.6% @ 167.5	/	/	17
	SPAN	35.1%	29.8% @ 837.5	/	/	18
	SPAN-Mo-475	31.4%	33.4% @ 500	/	/	19
SPAN	~45%	79.6% @ 600	/	/	20	

References

1. *ChemComm* **54**, 2288 (2018).
2. *Electrochim. Acta* **293**, 191 (2019).
3. *Mater. Lett.* **242**, 5 (2019).
4. *Small Methods* **6**, 2100899 (2022).
5. *ChemComm* **55**, 5267 (2019).
6. *J. Mater. Chem. A* **6**, 14587 (2018).
7. *ACS Appl. Energy Mater.* **5**, 11304 (2022).
8. *Batteries & Supercaps* **4**, (2021).
9. *J. Mater. Chem. A* **7**, 12732 (2019).
10. *Nano Letters* **13**, 4532 (2013).
11. *Adv. Funct. Mater.* **32**, 2201191 (2022).
12. *Adv. Energy Mater.* **12**, 2102836 (2022).
13. *Chem. Eng. J.* **426**, 130787 (2021).
14. *Energy Stor. Mater.* **23**, 8 (2019).
15. *ACS Appl. Mater. Interfaces* **14**, 6658 (2022).
16. *J. of Power Sources* **307**, 31 (2016).
17. *Electrochim. Acta* **333**, 135493 (2020).
18. *Adv. Energy Mater.* **11**, 2003469 (2021).

19. *Chem. Eng. J.* **404**, 126430 (2021).
20. *ACS Energy Lett.* **8**, 2746 (2023).

The sulfur contents of previous reported SPAN cathodes in Na/K-S batteries are mainly within the range of 25-42%. Based on these data, the sulfur content in $W_{SA}-W_2C@NC/S$ is actually on par or higher than those of SPAN cathodes in Na/K-S batteries. Even with similar or lower sulfur contents for previous studies, the sulfur utilization of SPAN cathodes in K-S batteries (26.9-83.6%) are still inferior to that of $W_{SA}-W_2C@NC/S$ (89.8%). Apart from sulfur utilization, the cyclability of SPAN cathodes along long-term cycling also cannot deliver any advantage over $W_{SA}-W_2C@NC/S$. As listed in Table R1, $W_{SA}-W_2C@NC/S$ delivered capacity retention of 56.8% upon 98 cycles at 1C, which is higher than most SPAN cathodes reported by far.

We thank the referee for mentioning the improving sulfur content in the composite. We have conducted experiments to increase the sulfur content in the composite. We believe that the sulfur content in the composite is largely determined by the porosity of the sulfur host. The total pore volume of $W_{SA}-W_2C@NC$ host is $0.315778 \text{ cm}^3 \text{ g}^{-1}$, which is provided by the carbon phase. This pore volume maximum accommodates 42.7 wt.% sulfur in the sulfur-host composite. The excess sulfur could only coat on the surface of the host, which would be removed by the 3rd step of sulfur impregnation procedure (200°C in flowing Ar). For example, we attempted to use higher sulfur : host ratio of 2:1 and 3:1 for the sulfur impregnation process. The final sulfur contents in these composites are displayed in Figure R3a. Due to the pore volume limit, increasing the original sulfur dosage cannot increase sulfur content in the obtained composite (Figure R3).

Figure R3. a TGA curves of sulfur/host composites using sulfur : host mass ratios of 1:1, 1:2 and 1:3. **b** The relationship between sulfur : host mass ratio and sulfur content in composite.

By deleting the 3rd step heating (200 °C in flowing Ar), the sulfur content in the composite can reach 49.46% if using sulfur : host ratio of 1:1. However, there is excess sulfur resides on host surface outside the porosity, which is detrimental for the K-S electrochemical performances (sulfur utilization of 76.5%, Figure R4a,b). If both deleting 3rd step heating and using sulfur : host ratio of 3 : 1, higher sulfur content of ~72.2% can be obtained. Due to too much sulfur that were not properly confined in the host, the sulfur utilization became lower (45.3%). For pursuing optimal performance in K-S batteries and for the purpose of systemically study the K-S electrochemistry mechanisms (highest sulfur utilization is needed), we adopted the current composite synthesis procedure.

Figure R4. a TGA curves of composites with different sulfur contents. **b** Voltage profiles of KSBs employing W_{SA}-W₂C@NC/S cathodes with different sulfur contents.

For sure we fully agree that pursuing very high sulfur content is an everlasting target for developing metal-sulfur batteries, which is also the ongoing studies in our groups. According to the current results, we discovered that the sulfur content can be further increased by creating larger pore volume in the host material without notably sacrificing electrochemical performance. These results are for future publications, but we can provide preliminary results of our next work upon request if was interested.

Referee comment #7:

The authors have mentioned that the first cathodic scan of $W_{SA}-W_2C@NC/S$, has a strong peak at 0.836 V corresponding to the SEI formation and the sulfur activation. Please add relevant references that confirm this statement. Also, why do the initial CV curves in Fig. S14 lack the peaks at R_1 and R_2 ? The reaction mechanism should be discussed correlating to this observation.

Author response #7:

We thank the referee for raising this important point. The shape of CV curve is indeed an important indicator for the reaction mechanism. Therefore, we fully agree with the referee that strong endorsements from relevant references should be included for the discussions of the CV curves about the K-S electrochemical reaction mechanism.

First, in the initial CV curve, the strong peak at 0.836 V is corresponding to the SEI formation and the sulfur activation. In the carbonate ester electrolyte for potassium based batteries, the SEI formation on the electrodes routinely occurred in the voltage region of 0.20-0.85 V vs. K/K⁺ (e.g. *Chem. Rev.* 122, 8053 (2022); *Adv. Energy Mater.* 12, 2103304 (2022)). SEI formation process in $W_{SA}-W_2C@NC/S$ cathode contributes current intensity in the broad 0.836 V cathodic peak. In addition, the sulfur activation is to describe the first potassiation process of sulfur confined in the host, which is the whole process from pristine sulfur to end member of K₂S. The redox current of the 1st sulfur potassiation process integrates into the broad 0.836 V cathodic peak.

Many high-impact references have mentioned this statement about the first cathodic peak in Na-/K-S batteries. We have attached more highly correlated references at the end of this statements in the revised manuscript.

Second, the shape difference between the 1st cycle CV and following cycle CV, *i.e.* the different position of cathodic peak, have been widely observed in previous Na-/K-S battery studies. Figure R5 presents the samples of this phenomenon in literature. In these studies, the 1st CV curves showed cathodic peaks at lower voltages than that in the following CV curves, making the 1st CV curves lacking the cathodic peaks of the following CV curves.

Figure R5. The CV curves profile of state-of-the-art K/Na-S batteries using carbonate ester electrolyte.

Combining our understanding and the opinions in literature, we state that the reason for this phenomenon is due to the different mechanisms of potassiating the pristine sulfur in the 1st cycle and potassiating the re-oxidized polysulfides in following cycles. The initial potassiation towards pristine sulfur (named sulfur activation) needs to overcome a higher energy barrier for taking place thereby delivering lower onset voltage in CV than the following potassiation towards polysulfides. Therefore, the redox peak located at lower voltage than that of R₁ and R₂ peaks. We supplemented discussions related to this point together with supporting references in the revised manuscript.

The text revisions are as below in blue font:

“.....

To understand the sulfur redox occurring in W_{SA}-W₂C@NC/S, W₂C@NC/S and NC/S cathodes, cyclic voltammety (CV) measurements were conducted. As shown in Supplementary Fig. 18, in the first cathodic scan of W_{SA}-W₂C@NC/S, there is a strong peak at 0.836 V combining the SEI formation^{9, 19, 22, 32} and sulfur activation processes.^{8, 47-49} The sulfur activation describes the potassiation of pristine sulfur confined in W_{SA}-W₂C@NC host, which probably needs to overcome higher energy barrier than the following potassiation processes.^{22, 44, 49} Therefore, the cathodic peak of the sulfur potassiation in the 1st cycle CV appeared at lower onset voltage than that in the following cycles.^{9, 19, 22}

.....”

References

- ACS Nano 13, 2536 (2019).

9. *Nanomaterials* **12**, 3968 (2022).
19. *Nat. Commun.* **12**, 6347 (2021).
22. *Adv. Mater.* **34**, e2106572 (2022).
32. *ACS Nano* **15**, 15218 (2021).
44. *Adv. Mater.* **34**, e2108363 (2022).
47. *Small Struct.* **3**, 2200020 (2022).
48. *ACS Nano* **15**, 20607 (2021).
49. *Carbon* **201**, 864 (2023).

Referee comment #8:

Additional discussions on the sulfur content in the $W_{SA}-W_2C@NC/S$, $W_2C@NC/S$, and NC/S composites based on the TGA curves should be added in the discussion part.

Author response #8:

We thank the referee for this good suggestion. Accordingly we have added more discussions about the sulfur content in the $W_{SA}-W_2C@NC/S$, $W_2C@NC/S$, and NC/S composites based on the TGA curves.

The relevant revision in the revised manuscript is as follow in blue font.

“.....

The $W_{SA}-W_2C@NC/S$ cathode delivered a high discharge capacity of 1504 mAh g⁻¹ based on the thermogravimetric analysis (TGA) determined sulfur content (40.56%) (Supplementary Fig. 19). This reversible capacity corresponds to 89.8% sulfur utilization, which is the highest value reported by far. The counterpart values for $W_2C@NC/S$ and NC/S are 82.8% and 73.3%, respectively, highlighting the advantage of $W_{SA}-W_2C$ composite in enhancing the electrochemical activity of sulfur. The discrepancy in sulfur content among three cathodes is below 3.4%, which is at the lowest level in the field (Supplementary Table 2). The cathodes with such sulfur content discrepancy deliver negligible difference in electrochemistry (Supplementary Fig. 20). The absolute sulfur contents above 40% are higher than that for reported sulfur-microporous carbon composites and on par with SPANs for KSBs (Supplementary Table 3).

.....”

New Supplementary Figure 20. **a** TGA data of $NC/S-New$, $W_2C@NC/S-New$ and $W_{SA}-W_2C@NC/S$. **b** GCD profiles of NC/S and $NC/S-New$. **c** GCD profiles of $W_2C@NC/S$ and $W_2C@NC/S-New$.

Referee comment #9:

Discussions on the type of nitrogen doping and its impact on the polysulfide conversion and electrochemical performances observed in the samples could be added.

Author response #9:

We thank the referee for raising this important point. To address this comment, we supplemented the analyses of nitrogen types and the discussions of the impact of nitrogen on electrochemical performances.

First, the types of nitrogen doping in NC , $W_2C@NC$, $W_{SA}-W_2C@NC$ were analyzed based on high resolution XPS N 1s spectra. As shown in Supplementary Fig. 22 and Supplementary Table 4 below, the nitrogen moieties in the three specimens all contain pyrrolic, pyridinic, graphitic and oxidized nitrogen. The ratios among these nitrogen species are highly close, indicating that the processes of anchoring tungsten species on NC have not affected the structures of nitrogen species.

New Supplementary Figure 22. High-resolution N 1s XPS spectra of **a** NC , **b** $W_2C@NC$ and **c** $W_{SA}-W_2C@NC$.

Supplementary Table 4. Nitrogen composition in different hosts according to high-resolution N 1s XPS spectra.

	$W_{SA}-W_2C@NC$	$W_2C@NC$	NC
Pyrrolic N	52.6%	51.1%	50.3%
Pyridinic N	32.0%	32.0%	32.3%
Graphitic N	9.6%	12.1%	10.1%
Oxidized N	5.8%	4.8%	7.3%

Second, based on the nearly identical compositions and structures of nitrogen doping in NC/S, $W_2C@NC/S$, $W_{SA}-W_2C@NC/S$, the difference in K-S electrochemical performance for the three cathodes should be essentially stemmed from the tungsten species. The reason is due to the much weaker interactions between nitrogen moieties and polysulfide than that between tungsten species and polysulfide. To prove this point, DFT calculations were first conducted to evaluate the adsorption of K_2S_4 on nitrogen or tungsten species. As shown in Table R2, the binding energy of K_2S_4 on W_2C in W_2C-W_{SA}/NC is much higher than that of K_2S_4 on the nitrogen site in W_2C-W_{SA}/NC and NC. Experimentally, polysulfide adsorption experiments were further conducted. Figure R6 displays the UV-vis spectra of K_2S_6 solutions soaked with NC, $W_2C@NC$, and $W_{SA}-W_2C@NC$ samples. Very weak K_2S_6 adsorption occurred on NC. On the contrary, the K_2S_6 solution became colorless after adding $W_{SA}-W_2C@NC$, suggesting the very strong adsorption of K_2S_6 by tungsten species. Therefore, it is safe to conclude that with the presence of tungsten species, the nitrogen doping is expected to have negligible impact on the KPSs adsorption and conversion.

Figure R6. UV-vis spectra and the optical photos (inset) of different K_2S_6 solutions with different host materials immersed after 12 h.

Table R2. The binding energies of K_2S_4 on W_2C in $W_{SA}-W_2C@NC$, nitrogen in $W_{SA}-W_2C@NC$, and nitrogen in NC.

K_2S_4 adsorption site	W_2C in $W_{SA}-W_2C@NC$	Nitrogen in $W_{SA}-W_2C@NC$	Nitrogen in NC
Binding Energy	-3.01 eV	-0.97 eV	-0.90 eV

The corresponding discussions in the revised manuscript are as below in blue font:

“.....

The extremely low capacities of the pure hosts (Supplementary Fig. 21) verified that the capacities of cathodes are derived from the reversible sulfur redox. Of note, the three specimens exhibited highly similar nitrogen moieties in terms of nitrogen types and relative ratios (Supplementary Fig. 22 and Supplementary Table 4). Therefore, the different electrochemical performances of three cathodes should be essentially stemmed from the tungsten species.

.....”

“**Electrochemical performances and sulfur redox mechanisms.** $W_{SA}-W_2C@NC/S$ utilizing the host containing functional species of W_{SA} and W_2C is expected to be high-perform cathodes for KSBs. First, $W_{SA}-W_2C@NC$ and $W_2C@NC$ demonstrated much stronger adsorption capability towards KPSs than that of tungsten-free NC, as proved by the ultraviolet-visible (UV-vis) spectra of K_2S_6 solutions with different host materials (Supplementary Fig. 17).^{45, 46} This phenomenon agrees well with the calculation demonstrated in Fig. 1.

.....”

References

45. *Carbon* **212**, 118173 (2023).
46. *Energy Stor. Mater.* **18**, 470 (2019).

Referee comment #10:

UV-visible spectroscopy is a simple and powerful technique to analyze the polysulfide adsorption and conversion capability of catalytic materials. Please add them to

confirm the adsorption and conversion capability of $W_{SA}-W_2C@NC$ and compare them with that of $W_2C@NC$ and NC.

Author response #10:

We thank the referee for the highly valuable suggestions. We fully agree that UV-visible spectroscopy is a powerful way to demonstrate the different behaviors of potassium polysulfide (KPSs) adsorption and conversion under the effects of different hosts. Therefore, we conducted supplementary experiments related to this point and included them in the revised manuscript.

First, to reveal the adsorption capabilities towards KPSs, we immersed the host materials of $W_{SA}-W_2C@NC$, $W_2C@NC$ and NC into K_2S_6 solutions. The UV-vis spectra of three solutions after 1 h and 12 h were shown in Supplementary Fig. 17. Clearly, the $W_{SA}-W_2C@NC$ host demonstrated the strongest adsorption capability towards K_2S_6 , as proved by the complete disappearance of the adsorbance peak at 290 nm ascribed to K_2S_6 . NC shows the weakest K_2S_6 adsorption due to the absence of tungsten species.

New Supplementary Figure 17. UV-vis spectra and the optical photos (inset) of different K_2S_6 solutions with different host materials immersed after **a** 1 h and **b** 12 h.

Second, to demonstrate the KPSs conversion catalyzed by different hosts, we assembled visual cells employing potassium foils and $W_{SA}-W_2C@NC/S$, $W_2C@NC/S$, NC/S as anodes and cathodes, respectively. As shown in Supplementary Fig. 31, the visual cells underwent 12 h standing and one entire discharging/charging cycle. The amount of KPSs dissolving in the electrolyte during this process is a reasonable descriptor for the conversion kinetics of KPSs. Because if the KPSs

transform efficiently, there is less chance for them to dissolve in the electrolyte. Per Supplementary Fig. 31, the electrolyte for $W_{SA}-W_2C@NC/S$ visual cell remain colorless during the whole process, while the electrolytes for $W_2C@NC/S$ and NC/S visual cells became yellow resulted from the KPS dissolving. The UV-vis spectra in Supplementary Fig. 32 display the difference more clearly. This phenomenon revealed by UV-visible spectroscopy and visual cells highlight the best catalytic capability of $W_{SA}-W_2C@NC$ towards KPSs conversion.

New Figure 4g. Visual cells under different states employing $W_{SA}-W_2C@NC/S$, $W_2C@NC/S$, NC/S cathodes and potassium foil anodes.

New Figure 4h. UV-vis spectra and the optical photos (inset) of visual cells employing different cathodes at the end of charge process.

The corresponding discussions in the revised manuscript are as below in blue font:

“**Electrochemical performances and sulfur redox mechanisms.** $W_{SA}-W_2C@NC/S$ utilizing the host containing functional species of W_{SA} and W_2C is expected to be high-perform cathodes for KSBs. First, $W_{SA}-W_2C@NC$ and $W_2C@NC$ demonstrated much stronger adsorption capability towards KPSs than that of tungsten-free NC , as

proved by the ultraviolet-visible (UV-vis) spectra of K_2S_6 solutions with different host materials (Supplementary Fig. 17).^{45, 46} This phenomenon agrees well with the calculation demonstrated in Fig. 1.

.....”

“.....

Of note, $W_2C@NC/S$ and NC/S delivered comparable high Tafel slopes near the O_1 peaks, which is direct evidence of the poisoning effect on W_2C catalytic site in $W_2C@NC/S$. The more facile KPSs conversion kinetics endowed by the synergy of W_{SA} and W_2C can also be demonstrated by visual cell measurements. As shown in Fig. 4g, the visual cells employing potassium foils as anodes and $W_{SA}-W_2C@NC/S$, $W_2C@NC/S$, NC/S as cathodes underwent 12 h standing and one entire discharging/charging cycle. The amount of KPSs dissolving in the electrolyte during this process is a reasonable descriptor for the KPSs conversion kinetics, because sluggish KPSs conversion provides more time for KPS dissolving. At the end point of charge-3.0 V, the electrolyte for $W_{SA}-W_2C@NC/S$ visual cell remained colourless with negligible KPS signal detected in UV-vis spectrum (Fig. 4h). On the contrary, the electrolytes for $W_2C@NC/S$ and NC/S visual cells dissolved much more KPS, suggesting the less facile KPS conversion.

.....”

Supporting Information

“.....

Adsorption measurements

The K_2S_6 solution for adsorption measurements was prepared by mixing potassium sulfide (K_2S) and sulfur with a molar ratio of 1:5 in dimethyl ether (DME). 5 mg $W_{SA}-W_2C@NC$, $W_2C@NC$, and NC were added into the 2 ml 0.02 M K_2S_6 solution, respectively, with the blank K_2S_6 solution as a reference.

Materials characterization

.....The ultraviolet-visible (UV-vis) spectra were performed by Shimadzu UV-2700 spectrophotometer.”

“.....

Visual cell measurements

The cathode materials ($W_{SA}-W_2C@NC/S$, $W_2C@NC/S$, or NC/S), MWCNTs, PVDF with a mass ratio of 6:3:1 was dispersed in NMP, and coated on carbon paper, followed by drying at 60 °C for overnight. The potassium foil that was stamped onto

the stainless-steel collector was used as anode. Both anode and cathode were clamped by alligator clips, and the electrolyte is 4 M KTFSI in DME. Finally, the reaction vessels were sealed for galvanostatic charge-discharge measurements at 0.1C on Solartron multi-channel electrochemical workstation.

.....”

References

45. *Carbon* **212**, 118173 (2023).
46. *Energy Stor. Mater.* **18**, 470 (2019).

Reviewer #2:

General comments:

This is a super-impressive and -creative research.

Potassium-based energy storage technologies are the cutting-edge field and solution for the large-scale low-cost stationary electrical energy storage systems (ESSs), and the potassium – sulfur (K-S) battery is the extremely significant focus. However, K-S batteries still suffer from many key challenges, including the very low utilization of sulfur, the sluggish conversion kinetics of potassium polysulfides (KPSs) and the resulting shuttling, as well as the difficult decomposition of solid discharge products. In this work, authors proposed an innovative strategy of coupling polysulfides migration and conversion enhancements for improving the potassium sulfur electrochemistry in battery. The role of polysulfides transportation in the K-S cell is for the first time emphasized and optimized. The concept of achieving dual-functionality by screening and constructing $W_{SA}-W_2C$ composite active site on sulfur host is enlightening. The performance of K^+ storage by sulfur cathode also presents new information for the community. Moreover, the thermodynamic and kinetic mechanisms of potassium sulfur reaction influenced by the host functionalities were clearly demonstrated based on comprehensive experiment and calculation results. Overall, the story falls in the scope of Nature communications and can bring new and inspiring ideas for the readers. Therefore, I strongly recommend it being accepted after a minor revision.

Author response:

We would sincerely like to thank the referee for positive endorsement for our manuscript. We also appreciate the insightful input and highly valuable suggestions. Rigorous revisions were conducted according to the detailed comments and suggestions raised by the referee. We believe that the supplement experimental tests and discussions greatly improved our work. All the revised or newly added contents have been marked in blue font, in both the Response Letter and revised manuscript.

Referee comment #1:

It is impressive to see the cyclability of the cells at such high sulfur utilization level. The authors are better to further investigate the resistance changes in the K-S cells for explaining the sulfur cathode durability enhancement.

Author response #1:

We sincerely thank the referee for this excellent suggestion. We supplemented detailed measurements on the resistance changes of the K-S cell employing W_{SA} - $W_2C@NC/S$, $W_2C@NC/S$ and NC/S cathodes and added the relevant discussions in the revised manuscript. The interface resistances were measured by electrochemical impedance spectra (EIS) at various cycles (fresh, 1st cycle, 5th cycle, 10th cycle, 50th cycle). The spectra were shown in Supplementary Fig. 26a-c. Supplementary Figure 26d and 26e displays the equivalent circuit for the spectra fitting. According to the resistance values obtained (Supplementary Table 5), the resistances of W_{SA} - $W_2C@NC/S$ cell are distinctly lower than those of NC/S . Specifically, before cycling, the KSBs employing the W_{SA} - $W_2C@NC/S$ cathode exhibited the lowest resistance (R_p , 1106 Ω) and the largest slope in the low-frequency Warburg diffusion range compared to the $W_2C@NC/S$ (1128 Ω) and NC/S (1722 Ω), indicating the fastest diffusion of potassium ions in W_{SA} - $W_2C@NC/S$ cathode. After cycling, KSBs with W_{SA} - $W_2C@NC/S$ cathode remained an extremely low CEI impedance (R_f) and charge transfer impedance (R_{ct}). More importantly, the resistance kept increasing during the cycling for NC/S and $W_2C@NC/S$ cells, which is in line with the faster decay in capacity upon cycling. The key to the superior performance of W_{SA} - $W_2C@NC/S$ over the control groups is the improved ionic diffusivity and electron conductivity of the host, which facilitates the surface diffusion and subsequent conversion of KPSs.

New Supplementary Figure 26. EIS spectra of KSBs with **a** $W_{SA}-W_2C@NC/S$, **b** $W_2C@NC/S$, **c** NC/S cathodes after different cycles at 0.1C and **d,e** corresponding equivalent circuit diagrams.

New Supplementary Table 5. EIS fitting results of KSBs with $W_{SA}-W_2C@NC/S$, $W_2C@NC/S$ and NC/S cathodes after different cycles at 0.1C.

Sample	$W_{SA}-W_2C@NC/S$				$W_2C@NC/S$				NC/S			
	R_s^a	R_p^b	R_f^c	R_{ct}^d	R_s	R_p	R_f	R_{ct}	R_s	R_p	R_f	R_{ct}
Fresh	7.80	1106	NA	NA	9.57	1128	NA	NA	5.82	1722	NA	NA
After 1 st cycle	6.01	NA	42.4	557	4.13	NA	41.9	770	5.65	NA	49.6	1319
After 5 th cycle	5.24	NA	47.5	576	3.45	NA	140	841	5.62	NA	53.3	1155
After 10 th cycle	4.86	NA	54.2	697	4.18	NA	272	1025	3.77	NA	72.4	2348
After 50 th cycle	5.08	NA	13.4	427	4.50	NA	309	731	4.72	NA	66.8	2448

^a R_s : Ohmic resistance. ^b R_p : Interface resistance for fresh cell. ^c R_f : Interface resistance by CEI. ^d R_{ct} : Charge transfer resistance.

In addition, the notable decrease of impedance after cycling was closely related to electrode activation and CEI formation compared to fresh KSBs. At the first 10 cycles, the gradual increase of R_f implies a gradual stabilization process of CEI.

The detailed texts revisions are in blue font as below.

“.....

Afterwards, the cathodes kept stable cycling for 200 and 97 cycles, resulting in capacity retention ratio of 49.4 and 56.8%. $W_{SA}-W_2C-H@NC/S$ and $W_{SA}-W_2C-L@NC/S$ showed inferior cyclability, again proving the optimal content of $W_{SA}-W_2C$ in $W_{SA}-W_2C@NC$. For comparison, $W_2C@NC/S$ cathode can last for only 127 and 40 cycles at 0.5 and 1C. The cell failure occurred even earlier for NC/S cathode, *i.e.* till 32th and 20th cycles. The cell failure can only be postponed by cycling the cathodes at very slow rate (Supplementary Fig. 25). The changes in resistances in the cells provide another aspect for reflecting the diversity in cyclability of the cathodes (details in Supplementary Fig. 26 and Table 5).

.....”

Supporting Information.

“Electrochemical impedance spectroscopy (EIS) was performed on KSBs with different cathodes over numerous cycling cycles to investigate the behavior of

electrode durability in more depth. As shown in Supplementary Fig. 24 and Supplementary Table 3, before cycling, the KSBs employing the $W_{SA}-W_2C@NC/S$ cathode exhibited the lowest resistance (R_p , 1106 Ω) and the largest slope in the low-frequency Warburg diffusion range compared to the $W_2C@NC/S$ (1128 Ω) and NC/S (1722 Ω), indicating the fastest diffusion of potassium ions in $W_{SA}-W_2C@NC/S$ cathode. After cycling at 0.1C, KSBs with $W_{SA}-W_2C@NC/S$ cathode remained an extremely low CEI impedance (R_f) and charge transfer impedance (R_{ct}). The resistance kept increasing during the cycling for NC/S and $W_2C@NC/S$ cells, which is in line with the faster decay in capacity upon cycling. Therefore, the key to the superior performance of $W_{SA}-W_2C@NC/S$ over the control groups is the improved ionic diffusivity and electron conductivity of the host, which facilitates the surface diffusion and subsequent conversion of KPSs. In particular, the $W_{SA}-W_2C@NC/S$ cathode maintained a low impedance (427 Ω) after 50 cycles with $W_2C@NC/S$ and NC/S as high as 731 Ω and 2448 Ω , respectively, which means that hosts with $W_{SA}-W_2C$ sites did not have a significant accumulation of inactivated solid-phase KPSs.”

Referee comment #2:

I notice that normally one transition state between initial and final states was determined for calculating the energy barriers (figure 1b, 4e, 5f). For more accurate calculation, two or more transition states should be considered in order to obtain accurate energy profiles.

Author response #2:

We fully agree with the referee that more transition states should be considered to obtain more accurate energy profiles. In fact, we have considered several transition states in the calculation of energy profile for each process. We only showed the key transition state in original figures, which determined the energy barriers of the processes. To address this comment, we added more energy data points of transition states in each profile. As we can see, the additional transition state points don't change the energy barriers. The new Figure 1b, 4e and Supplementary Figure 32 are as below.

Fig. 1 Theoretical guidance and screening for sulfur host design. b The energy profiles of K_2S_2 migration on NC, $W_{SA}@NC$, and W_2C (102).

Fig. 4 Sulfur redox kinetics mechanism under synergy strategy. e Energy profiles for K_2S dissociation on $W_{SA}-W_2C@NC$, $W_2C@NC$ and NC.

Supplementary Figure 32. The initial, transition, final configurations of K_2S and K_2S_2 migration from W_2C catalytic site to the substrates for $W_2C@NC$ and $W_{SA}-W_2C@NC$ hosts.

Referee comment #3:

The design of $W_{SA}-W_2C$ composite is interesting. The EXAFS data in figure 1d proved the existence of single W atoms, but it will be helpful for deep structure-property correlation analysis if the authors can further determine the accurate configuration.

Author response #3:

We thank the referee for raising this important point. To address this comment, we attempted to fit the W L-edge EXAFS of $W_{SA}-W_2C@NC$. First, we constructed configuration model based on systematic theoretical screening. Second, we fitted the EXAFS spectrum to identify the accurate coordination environment of tungsten in the specimen. As shown in Figure R7, the model of $W_{SA}-N-C$ isolated W_2C cluster can well describe the atomic configuration of $W_{SA}-W_2C$ composite, as verified by the excellent fitting of EXAFS spectrum.

Figure R7. EXAFS experimental data and fitting curves for $W_{SA}-W_2C@NC$.

Table R3. EXAFS fitting parameters at the W L-edge for $W_{SA}-W_2C@NC$.

Path	C.N.	S_0^2	R (Å)	$\sigma^2 \times 10^{-3}$ (Å ²)	ΔE (eV)	R factor
W-C	5.8	0.90	2.91	1.0	-0.304	0.11
W-W	5.8		2.72	5.3		
W-N	4.0		1.79	16.9		

We also state that EXAFS fitting can serve as reference to understand the neighbor environments of W. According to the fitting result (Table R3), there are indeed nitrogen ligand for W, and also W-C bonding ascribed to tungsten carbide

Referee comment #4:

As supplemental analysis for the polysulfide conversion and the possible shuttling, the characteristics of the cycled potassium anode in different cells should be further revealed.

Author response #4:

The referee has raised a very important point. The characteristics of post-cycled potassium anode provide an important perspective for analyzing the shuttling effect of sulfur cathode. Therefore, we collected the potassium metal foils from the cell after cycling measurements and made comparative analyses.

Figure R8. Digital photos of corresponding potassium anodes in KSBs employing NC/S, W₂C@NC/S and W_{SA}-W₂C@NC/S cathodes after 98 cycles.

Figure R8 displayed the potassium foils collected from different cells, inhomogeneous potassium deposition can be observed on the K-foils in KSBs applying NC/S cathodes, suggesting the severely unstable potassium deposition caused by the polysulfide shuttling. This phenomenon can be more clearly observed in SEM images (Figure R9). The K foil collected from the W₂C@NC/S based cell was relatively flatter than the K foil from NC/S based cell, nonetheless there were also severe dendrites and bubbles in the cracks. The K foil W_{SA}-W₂C@NC W_{SA}-W₂C@NC/S based cell is extremely flat. Negligible dendrites and cracks were observed in the SEM image, indicating the stable reversible potassium deposition and stripping.

More direct evidence of polysulfide shuttling is the distribution of potassium and sulfur on the surface of cycled K foil. As demonstrated in Figure R9, strong sulfur

signal can be detected from the K foil from NC/S based cell, indicating the severe KPS accumulation on K foil due to KPS shuttling from cathode side. The K foil from $W_2C@NC/S$ based cell demonstrated the similar phenomenon due to the long cycling. On the contrary, very weak sulfur signal can be detected from the potassium foils collected from $W_{SA}-W_2C@NC/S$ based cell, suggesting the well confinement of KPSs in the cathode side and the effectively restricted KPSs shuttling.

Basically, the little KPS shuttling effect in the $W_{SA}-W_2C@NC/S$ strongly proves the highest polysulfide conversion kinetics, which highlights the critical role of W_{SA} and W_2C synergy in enhancing K-S electrochemical performance.

Figure R9. Morphologies and elemental maps of potassium anode in KSBs with **a-d** NC/S, **e-h** $W_2C@NC/S$, and **i-l** $W_{SA}-W_2C@NC/S$ cathodes.

Referee comment #5:

The authors are suggested to consider more control samples to verify the unique effect of $W_{SA}-W_2C$ composite, for instance the single W atom based host containing no W_2C .

Author response #5:

We thank the referee for this instructive suggestion. We fully agree with the referee that the W_2C -free single W atom based sulfur host should be considered as a control baseline. To address this comment, we synthesized $W_{SA}@NC$ host material. Following the same sulfur impregnation procedure as other specimens, $W_{SA}@NC/S$ was prepared and employed as cathode for K-S cell measurements.

The characterization results of $W_{SA}@NC$ and $W_{SA}@NC/S$ were shown in Figure R10. The morphology of $W_{SA}@NC$ was similar to that of NC. Both HRTEM and SAED showed the characteristics of carbon materials. In the HADDF STEM images, the presence of W single atoms can be clearly distinguished, which is consistent with the W signal shown by EDS mapping. After three-step sulfur impregnation, the morphology of $W_{SA}-W_2C@NC/S$ demonstrate negligible change compared to $W_{SA}@NC$. The 39.55 wt.% sulfur content in $W_{SA}@NC/S$ can be confirmed by TGA. Moreover, the micropores of $W_{SA}@NC$ were preferentially filled with elemental sulfur, which exists in amorphous texture.

Figure R10. Characterizations of $W_{SA}@NC$ and $W_{SA}@NC/S$. **a-b** HRTEM images, **c** HADDF-STEM image and **d** EDS mapping of $W_{SA}@NC$. **e-f** HRTEM images, **g** SAED pattern and **h** EDS mapping of $W_{SA}@NC/S$. **i** TGA curve of $W_{SA}@NC/S$. **j** Nitrogen adsorption-desorption isotherms, **k** pore distribution and **l** XRD patterns of $W_{SA}@NC$ and $W_{SA}@NC/S$.

The charge/discharge profiles, the rate performance and cyclability of $W_{SA}@NC/S$ were demonstrated in Figure R11, which are all inferior to that of $W_{SA}-W_2C@NC/S$ cathode. In particular, the $W_{SA}@NC/S$ cathode delivered reversible capacity of 1307 mAh g^{-1} at $0.1C$ leading into sulfur utilization of ca. 78%, which is lower than that of $W_{SA}-W_2C@NC/S$ cathode. In addition, the discharge capacities of the $W_{SA}@NC/S$ cathode were 1304, 960, 819, 674, 508, 327, and 189 mAh g^{-1} at

current densities of 0.1C, 0.2C, 0.3C, 0.5C, 1C, 2C, and 3C. Compared to the failure of the $W_2C@NC/S$ at high current densities, the longer lifespan of the $W_{SA}@NC/S$ was mainly derived from the effect of W_{SA} on the assisted migration of KPSs in the active site. Moreover, the cyclabilities of $W_{SA}@NC/S$ cathode at various rates are all worse than those of $W_{SA}-W_2C@NC/S$ cathode. This control experiment verified the significance of the synergy between W_{SA} and W_2C species in enhancing K-S electrochemistry performance.

Figure R11. Electrochemical performances of $W_{SA}@NC/S$ cathode. **a** GCD profiles curves at 0.1C, **b** rate-performance and **c,d** cyclability of KSBs employing $W_{SA}@NC/S$ cathode at 0.1C (**c**) and 0.5C (**d**).

Reviewer #3:**General comments:**

The work reported by W. Song et al. used W_{SA} - W_2C -NC composite as a sulfur for improving the performance of KSBs. The theoretical approach has been adopted to synthesize the composite for better electrochemical features. Further, a detailed analysis has been carried out to investigate the role of W_{SA} - W_2C in polysulfide capture and conversion. After careful evaluation, the manuscript can be improved further with a major revision before acceptance for the publication. Specific comments are provided below:

Author response:

We sincerely appreciate the reviewer for the careful reading of our work. We would like to thank referee for the insightful inputs and inspiring suggestions. We provide point-to-point responses to the comments raised by the referee. Comprehensive experiments and credible discussions were supplemented in the revised manuscript. We believe the quality of the work is much improved as a result of the constructive additions. The revised texts and figures are in blue font in both the Response Letter and revised manuscript.

Referee comment #1:

Introduction should be modified to illustrate why W_{SA} , W_2C , and N-doped carbon have been adopted for this work.

Author response #1:

We sincerely thank the referee for this excellent suggestion. We fully agree that the motivation of adopting tungsten species of W_{SA} , W_2C and N-doped carbon should be illustrated in the Introduction. To address this comment, we revised the Introduction discussing the advantages of tungsten based species and N-doped carbon. The revisions are as below in blue font.

Introduction

“.....

Some pioneering efforts were made to innovate functional sulfur hosts for KSBs. Most of the hosts are carbon-based, taking advantage of the good electrical

conductivity, diverse surface chemistry, and versatile porosity of carbon.^{6,7} The sulfur confinement is the primary functionality of the host. Physically confining sulfur in the carbon porosity is a popular strategy.^{8,9} For instance, microporous carbons confining small molecular (S_{1-3}) exhibited promising capacity and cyclability in KSBs. Nonetheless, the limitation of pore volume normally sacrificed the sulfur loading.^{7,10} The carbon hosts employed were routinely functionalized by nitrogen moieties, which are effective of enhancing the sulfur confinement via covalent bonding at heterointerfaces.¹¹ In addition, the chemical confinement strategy was also widely reported.¹²⁻¹⁴ Representatively, sulfurized polyacrylonitrile (SPAN) utilizing the covalent sulfur-carbon bonds greatly inhibited KPSs shuttling.¹⁴⁻¹⁷ The chemical confinement also includes the chemical adsorption between KPSs and certain polar species of the host, such as heteroatom groups, nanostructured compounds, atomic metal assemblies.^{7,18,19} For instance, single metal atoms^{18,20,21} and metal carbides²²⁻²⁵ were reported providing adsorption capability towards polysulfides.

The functionality of catalytic conversion towards KPSs for sulfur host attracted increasing attention in recent years. Some highly enlightening studies introduced catalytic sites in the hosts in order to improve the kinetics of KPSs conversion, especially the solid-state reactions like K_2S oxidation.²⁶ Metallic single atoms^{18,26} and metallic atom clusters²⁷ were reported as electrocatalytic species in the hosts, which suppressed the long-chain KPSs shuttling and reduced the polarization of solid-state sulfide conversion, thereby enhancing the sulfur utilization, rate and cycling capability. The design of catalytic materials for KPSs can be inspired by the prior achievements in Li/Na-sulfur batteries. The metal element of various forms (*e.g.* single atom, atomic cluster, compounds) that can deliver general catalytic capability towards both lithium and sodium polysulfides deserves particular attention. For example, tungsten can be catalytically active in Li-S batteries (tungsten single atom²⁰, tungsten carbide²³⁻²⁵, tungsten sulfide^{28, 29}) and Na-S batteries (tungsten nanoparticle³⁰). The successful applications of tungsten-based catalysts in metal sulfur batteries were largely ascribed to the special electronic structure of tungsten atomic assemblies and compounds.³¹ These insights are highly instructive for current KSBs explorations.

.....”

References

11. *Adv. Energy Mater.* **10**, 2000931 (2020).

18. *Angew. Chem. Int. Ed.* **62**, e202301681 (2023).
20. *Angew. Chem. Int. Ed.* **60**, 15563 (2021).
21. *Adv. Mater.* **35**, 2208873 (2022).
22. *Adv. Mater.* **34**, e2106572 (2022).
23. *Nano Energy* **59**, 636 (2019).
24. *Chem. Eur. J.* **26**, 16057 (2020).
25. *Adv. Mater. Interfaces* **6**, 1802088 (2019).
28. *Adv. Energy Mater.* **10**, 2000091 (2020).
29. *Adv. Energy Mater.* **7**, 1601843 (2017).
30. *Adv. Sci.* **9**, 2105544 (2022).

Referee comment #2:

For sulfur composite preparation: (a) Why 1:1 mass ratio is used? (b) Why has three-step heating been followed?

Author response #2:

We appreciate the referee for raising this very good point. The initial ratio of host and sulfur for preparing the sulfur composite is indeed a key parameter affecting the cathode performance. The ratio of 1:1 is the optimal value obtained by conducting systematic controlling experiments. The reason for adopting three-step heating during the sulfur impregnation process is for pursuing the best K-S electrochemical performance.

(a) We have conducted control experiments to determine the best sulfur : host mass ratio. We have chosen mass ratios of 1:1, 2:1 and 3:1, and proceeded the identical three-step heating process. We discovered that adopting higher sulfur : host mass ratio cannot proportionally increase the sulfur content in the obtained composites (Figure R12). The reasons for the saturation of sulfur content in the host are ascribed to the porosity structure of the host and the three-step heating procedure. Given the pore volume of $0.315778 \text{ g cm}^{-3}$ for $\text{W}_{\text{SA}}\text{-W}_2\text{C@NC}$, theoretically the highest sulfur content in the S/C composite is 42.7 wt.% if assuming the porosity were fully filled by sulfur. Moreover, the three-step heating procedure can thoroughly remove the sulfur residue outside the porosity, leading to the ceiling of sulfur content in the composites.

Figure R12. **a** TGA curves of sulfur/host composites using sulfur : host mass ratios of 1:1, 2:1 and 3:1. **b** The relationship between sulfur : host mass ratio and sulfur content in composite.

(b) The reason for following three-step heating is because it is the best procedure for impregnating sulfur into the porosity of the host. Due to the tortuosity of host porosity and surface tension of melted sulfur, the heating at 155 °C (1st step) in sealed vacuum tube cannot fully infuse sulfur into the porosity. As shown in the SEM image of Figure R13a, there are sulfur particles remaining in the carbon surface. The 2nd heating step was carried out at 300 °C, when sulfur was transformed into the gaseous state at the laboratory vacuum level of $\sim 10^{-2}$ atm. Driven by saturated vapor pressure, more gaseous sulfur species were pushed into the porosity (Figure R13b). The excess sulfur on the host surface after 2nd step can be further removed after heating at 200 °C under flowing argon (3rd step), as proved by the SEM image of Figure R13c.

Figure R13. Structure and morphology characterization of carbon/sulfur composites under different heating treatment. **a-c** SEM images and corresponding schematic illustrations of composites after 1st step (a), after 2nd step (b), and after 3rd step (c) heating treatment.

The three-step heating procedure was adopted for sulfur impregnation because it ensured the best K-S electrochemical performance. Sulfur residues on the host surface delivered much inferior activity and reversibility in KSBs, which can be thoroughly remove by three-step heating. We also prepared composites following one-step heating and two-step heating (1:1 mass ratio), which have sulfur contents of 49.53% and 49.46% respectively (Figure R14a). As shown in Figure R14b and R14c, the three-step heating specimen delivers the highest sulfur utilization and the lowest discharge/charge polarization. Moreover, the practical capacity approaching theoretical value for sulfur cathode is also highly important for the mechanism study for K-S electrochemistry because full sulfur utilization provides all the reaction intermediates and products for analyses.

Figure R14. **a** TGA curves of composites following one-/two-/three-step heating. **b** Voltage profiles of the three composites. **c** Corresponding sulfur utilization.

Referee comment #3:

For electrode preparation: (a) Why was composite only 60% while 40% (CNT and binder) has been used, as this decreases the active material on the electrode? Currently, sulfur content on the electrode is only $0.6 \times 0.4 = 0.24$ fraction (=24%). (b) What is the sulfur loading of the electrode?

Author response #3:

We thank the referee for raising the important points about the mass ratio and the sulfur loading of the electrode. We provide the following data and discussions to address this comment.

(a) The optimized ratio of 6:3:1 was obtained by conducting control experiments. The use of higher amount (30%) CNT endows the cathode with best electrochemical performance in terms of gravimetric capacity, areal capacity, sulfur utilization, polarization and cyclability. We have tested the electrodes with mass ratio of 7:2:1 and 5:4:1. As shown in Figure R15, the electrode with ratio of 7:2:1 delivered lower sulfur utilization of 75.3%. Of note, the increase of CNT content to 40% cannot further improve the reversible capacity as compared to the 30% CNT content counterpart. Therefore, 6:3:1 is the optimal ratio for preparing electrodes. Moreover, approaching theoretical capacity for sulfur cathode is also highly important for the mechanism study for K-S electrochemistry because full sulfur utilization provides all the reaction intermediates and products for analyses.

Figure R15. GCD profiles of electrodes prepared with different active material : CNT : binder ratios.

We believe the reason for requiring 30 wt.% CNT in the electrode is largely due to the not so good conductivity of CNT in our lab. We tried to optimize the electrode structure in order to address this drawback in conductive carbon, and afterwards the mass ratio of CNT can be effectively reduced. For instance, we coated slurry comprising 70% active materials, 20% CNT, and 10% binder on a carbon paper current collector instead of on Al foil. As shown in Figure R16, this electrode delivered a very high sulfur utilization at higher sulfur loading (95% with $0.94 \text{ mg}_{\text{sulfur}} \text{ cm}^{-2}$ at 0.1C).

Figure R16. GCD profiles of electrode prepared with carbon paper collector and 20% CNTs.

(b) The sulfur loading of the electrode for routine electrochemistry measurements is *ca.* 1 mg cm⁻². According to Supplementary Table 3, the sulfur loading of ~1 mg cm⁻² is a widely applied parameter in the previous research of Na-S and K-S batteries in the field, which is also a proper sulfur loading level for studying the Na-S/K-S electrochemistry.

For W_{SA}-W₂C@NC/S based cathode, we can also increase the sulfur loading over 2.5 mg cm⁻², which has never been reported in K-S battery system to date. As shown in Supplementary Fig. 23, even in the high sulfur loading of 2.86 and 3.61 mg cm⁻², the sulfur utilization can still reach 88.2% and 72.9%, which still have advantage over the state-of-the-art results under low sulfur loading conditions.

New Supplementary Table 3. Comparison between W_{SA}-W₂C@NC/S and the state-of-the-art cathodes for KSBs.

Cathodes	Sulfur content (%)	Sulfur loading (mg cm ⁻²)	Maximum Sulfur utilization	Rate performance (Capacity (mAh g ⁻¹) @ Current (mA g ⁻¹))	Ref.
W _{SA} -W ₂ C@NC/S	40.56%	~1.0/ ~2.8	89.8%/ 88.2%	1504 @ 167.5 1214 @ 837.5 1059 @ 1675	This work
C/S composite	18.6%	0.5–1.0	77.2%	1293.3 @ 20 741.2 @ 2000	10
PCNF/S	25%	0.5–1.0	83.0%	1392 @ 20 1138 @ 200	11
S@P-NCF	37%	~0.8	87.7%	1470 @ 337 560 @ 3370	12
SPAN	39.52%	Not reported	31.7%	532 @ 35	13

SPAN	38%	~1.0	42.4%	710 @ 47.5 218 @ 285	14
SPAN	36%	~0.8	58.9%	987 @ 837.5	15
I-S@pPAN	~42%	~1.0	86.1%	1442 @ 167.5	16
FS-SPAN	33%	~0.8	80.2%	1345 @ 70 680 @ 1000	17
γ S-CNFs	~50%	0.69	52.8%	885 @ 167.5 66 @ 3350	18
CCS	39.25%	0.39–0.59	26.9%	440 @ 150 94 @ 1000	19
EAMC12	37.8%	1.5–2.0	69.5%	1165 @ 335 572 @ 1675	20
S@SA-NC	51.2%	~1.0	75.6%	1266 @ 337 868 @ 837.5	21
CCS@CBC_450 (3 M KTFSI in TEGDME)	40%	~1.5	42.4	711 @ 100 130 @ 1000	22
S/CNF (1 M KCF ₃ SO ₃ in TEGDME)	27.3%	~1.0	67.2%	1126 @ 167.5 938 @ 335 780 @ 558.3	23
CNT/S (3 M KFSI in DME)	25.5%	~0.2	18.6%	311 @ 50 94 @ 500	24
ACF-1500@S (3 M KFSI in DME)	15%	~1.3	18.1%	303 @ 50	25
S ₈ /VC (0.3 M Cu(TFSI) ₂ -0.1 M KTFSI in Me-Im)	Not reported	1–1.2	50.6%	847 @ 150	26
0.05 M K ₂ S ₅₋₆ catholyte (0.5 M KTFSI in DEGDME)	Not reported	Not reported	22.3%	374 @ 55.8 107 @ 1116	27
0.02 M K ₂ S ₅ catholyte (1 M KTFSI in DEGDME)	Not reported	~0.128	33.2%	556 @ 117	28

References

- ACS Nano **13**, 2536 (2019).
- J. Mater. Chem. A **8**, 10875 (2020).
- Nanomaterials **12**, 3968 (2022).
- Mater. Lett. **242**, 5 (2019).
- Chem. Commun. **54**, 2288 (2018).
- J. Mater. Chem. A **6**, 14587 (2018).
- Chem. Commun. **55**, 5267 (2019).
- Small Methods **6**, 2100899 (2022).
- J. Mater. Chem. A **11**, 15924 (2023).
- Electrochim. Acta **293**, 191 (2019).
- Chem. Commun. **57**, 1490 (2021).

21. *J. Am. Chem. Soc.* **143**, 16902 (2021).
22. *ACS Sustain. Chem. Eng.* **10**, 16634 (2022).
23. *Energy Stor. Mater.* **15**, 368 (2018).
24. *J. Electron. Mater.* **50**, 3037 (2021).
25. *J. Power Sources* **480**, 228874 (2020).
26. *J. Mater. Chem. A* **7**, 20584 (2019).
27. *ACS Energy Lett.* **3**, 540 (2018).
28. *Angew. Chem. Int. Ed.* **61**, e202200606 (2022).

New Supplementary Figure 23. **a** GCD profiles curves at 0.1C and **b** cyclability of KSBs employing $W_{SA}-W_2C@NC/S$ cathodes with high sulfur loadings.

The relevant revision in text in the revised manuscript are as below in blue font.

“.....

The extremely low capacities of the pure hosts (Supplementary Fig. 21) verified that the capacities of cathodes are derived from the reversible sulfur redox..... $W_{SA}-W_2C@NC/S$ electrodes with high sulfur loading of 2.86 and 3.61 $mg\ cm^{-2}$ delivered slightly lower capacities of 1477.8 and 1221.9 $mAh\ g^{-1}$ (Supplementary Fig. 23), which are still higher than most previously reported cathodes with low sulfur loading (Supplementary Table 3).

.....”

Supporting Information.

“.....

The electrolyte is 0.8 M KPF_6 in EC/DEC with a volume ratio of 1:1. The typical sulfur loading of the electrode is ca. 1 $mg\ cm^{-2}$. Electrodes with high sulfur loading of ca. 2.8 and ca. 3.6 $mg\ cm^{-2}$ were also prepared for electrochemical tests. For the full wetting of the glass fiber separator, the normal electrolyte dosage in one coin cell is 100 μL .

.....”

Referee comment #4:

Infusion of sulfur is known to increase the interlayer spacing. In this case, why does it not affect the interlayer spacing of W₂S?

Author response #4:

We thank the referee for raising this very interesting point. To address this comment, we have carefully analyzed the possibility of sulfur infusion into the interlayer of W₂C and the effect on W₂C interlayer spacing (there is no W₂S phase in all our materials, thus we think the referee was probably meaning W₂C).

First, W₂C is not a layered phase. The following Table R4 lists the interlayer spacing of the main crystal planes (002), (200) and (102) of W₂C, all of which are much smaller than the interlayer spacing of the (002) plane of layered WS₂ phase. According to the atomic configuration of S₂₋₈ shown in Figure R17, we believe that even the smallest S₂ molecule is thermodynamically unlikely to enter the interlayer of W₂C due to the huge steric hindrance.

Table R4. Comparison of the interlayer spacing between WS₂ (JCPDS No. 08-0237) and W₂C (JCPDS No. 20-1315) main crystal planes.

WS ₂	Interplanar spacing (Å)	W ₂ C	Interplanar spacing (Å)
(002)	6.18 Å	(002)	2.60 Å
(004)	3.09 Å	(200)	2.36 Å
(100)	2.73 Å	(102)	2.28 Å

Figure R17. The sulfur allotropes from S_2 to S_8 obtained by density function theory (DFT) calculation. (Cited from: *Journal of the American Chemical Society* 134, 18510 (2012).)

Second, in order to further confirm whether there is any sulfur insertion into the interlayers of W_2C in the sulfur-carbon composites, EDS line sweeps were used to perform elemental analyses on W_2C nanocrystals in $W_{SA}-W_2C@NC/S$. As shown in Figure R18, the change trends of sulfur and tungsten signal intensity are quite different. The intensity of sulfur signal remains constant within and outside the particle regions, indicating the absence of sulfur enrichment in the particle. On the contrary, the tungsten signal intensity fits the shape of the particle. Therefore, no sulfur enrichment could be observed on randomly selected W_2C nanoparticles, which objects to the sulfur infusion in W_2C .

Figure R18. EDS line scan of W and S elements across W_2C nanocrystals in $W_{SA}-W_2C@NC/S$.

Furthermore, we measured the lattice spacing of W_2C nanoparticles as shown in Figure R19. The lattice spacing of W_2C nanoparticles did not change before and after sulfur infusion, proving that sulfur molecules did not enter the W_2C interlayer.

Figure R19. Lattice spacing of W_2C nanoparticles on $W_{SA}-W_2C@NC$ and $W_{SA}-W_2C@NC/S$.

Referee comment #5:

The sulfur composites have been synthesized at the melting point of sulfur. How do the composites show the amorphous nature of sulfur?

Author response #5:

We thank the referee for the careful reading of our manuscript and the inspiring question. Referee mentioned a very important point about the nature of sulfur existing in the composite. For synthesizing host-sulfur composites, we adopt three-step heating procedure. The temperatures of the three-step heating are respectively 155 °C (in sealed vacuum glass tube), 300 °C (in sealed vacuum glass tube) and 200 °C (in flowing Ar gas), which are all above the melting point of sulfur (112 °C).

In order to explain how the amorphous sulfur forms after the three-step heating procedure, we characterized the sulfur texture after each heating step (Figure R20). After the first heating step (155 °C for 24 h at sealed vacuum tube), sulfur is in the orthorhombic phase. The orthorhombic sulfur exists mainly on host surface, as revealed by the SEM image (Figure R20a and R21). The second heating step at 300 °C resulted in significant decrease in the crystallinity of sulfur, whose XRD pattern showed diffraction peaks of monoclinic phases. According to the phase diagram of sulfur (Figure R21), the transformation from point ① to ② can be realized in the second heating step under laboratory level vacuum condition (*ca.* 10^{-2} atm). Sulfur evaporated during the second heating step. The vapor sulfur breaks the barrier of liquid sulfur restricted by surface tension and can infiltrate more into the micropores of the host material. In the process of cooling to room temperature, part of the sulfur that does not restrict by the micropores of the host (Figure R20c) transformed into the monoclinic phase.

Figure R20. Structure and morphology characterization of carbon/sulfur composites under different heating treatment. **a** XRD patterns and **b-d** SEM images and corresponding schematic illustration of composites under one-step (b), two-step (c), and three-step (d) heating treatment.

Figure R21. Phase diagram of sulfur. (Reproduced from: *The Journal of Chemical Thermodynamics* 43, 1224 (2011).)

After the third heating step for removing superficial sulfur, the signal of the crystalline sulfur disappears (as shown in Figure R20d), indicating the amorphous texture of sulfur remaining in the composite. One reason for the amorphous nature of sulfur after three-step heating procedure is the thorough removing of crystalline sulfur on host surface. Another more important reason is that the sulfur species being confined in the microporosity (0.68 nm in diameter for NC, Supplementary Fig. 16b) cannot form long-range order crystallinity due to the steric hindrance of porosity. As shown in Figure R17, only small molecular sulfur species without long range-order can exist within the microporosity, thereby delivering amorphous nature.

The amorphous sulfur accommodated in microporosity of hosts were frequently reported in the metal-sulfur battery community. The following Table R5 listed many examples of host/sulfur composites synthesized by similar procedures.

Table R5. Examples of host/sulfur composite, sulfur impregnation procedure, XRD patterns collected from previous literatures of Na/K-sulfur batteries.

Sample	Sulfur impregnation procedure	XRD patterns	Ref.
W_{SA^-} $W_2C@NC/S$	155 °C for 24 h in vacuum; 300 °C for 2 h in vacuum; 200 °C for 1 h in Ar		This work

PCNF/S	500 °C for 5 h in vacuum; 200 °C for 4 h in Ar		Nanomaterials 2022, 12, 3968
Microporous C/S composite	600 °C for 5 h in vacuum; 200 °C for 5 h in Ar		ACS Nano 2019, 13, 2536
MMPCS- 800@S	155 °C for 12 h in vacuum; 300 °C for 2 h in vacuum		Adv. Mater. 2022, 34, 2108363
MoC/Mo ₂ C@P CNT-S	155 °C for 12 h in vacuum; 300 °C for 2 h in vacuum		Adv. Mater. 2022, 34, 2106572
Co@PCNFs/S	300 °C for 12 h in vacuum		Adv. Funct. Mater. 2021, 31, 2102280
CN/Au/S microspheres	155 °C for 12 h in vacuum; 350 °C for 2 h in vacuum		Energy Environ. Sci. 2020, 13,562
ACC-xS	155 °C for 15 h in vacuum; 200 °C for 20 min in Ar		Adv. Sci. 2020, 7, 1903246

Referee comment #6:

The electrochemical performance should be performed with higher sulfur loading (> 2.5 mg cm⁻²).

Author response #6:

We fully agree with the referee that it is highly instructive to test the high loading sulfur cathodes. We added supplementary experiments to demonstrate the electrochemical performance of high sulfur loading $W_{SA}-W_2C@NC/S$ cathodes. The sulfur loading of the cathodes tested are all above 2.5 mg cm^{-2} .

New Supplementary Figure 23. a GCD profiles curves at 0.1C and **b** cyclability of KSBs employing $W_{SA}-W_2C@NC/S$ cathodes with high sulfur loadings.

Supplementary Figure 23 displayed the charge profiles of the $W_{SA}-W_2C@NC/S$ cathodes with high sulfur loading. At sulfur loading of 2.86 and 3.61 mg cm^{-2} , the specific capacities obtained are 1477.8 and 1221.9 mAh g⁻¹, corresponding to sulfur utilization of 88.2% and 72.9%.

The relevant text revisions in the revised manuscript are as followed in blue font.

“.....

The extremely low capacities of the pure hosts (Supplementary Fig. 21) verified that the capacities of cathodes are derived from the reversible sulfur redox..... $W_{SA}-W_2C@NC/S$ electrodes with high sulfur loading of 2.86 and 3.61 mg cm^{-2} delivered slightly lower capacities of 1477.8 and 1221.9 mAh g⁻¹ (Supplementary Fig. 23), which are still higher than most previously reported cathodes with low sulfur loading (Supplementary Table 3).

.....”

Supporting Information.

“.....

The electrolyte is 0.8 M KPF₆ in EC/DEC with a volume ratio of 1:1. The typical sulfur loading of the electrode is ca. 1 mg cm⁻². Electrodes with high sulfur loading of ca. 2.8 and ca. 3.6 mg cm⁻² were also prepared for electrochemical tests. For the full wetting of the glass fiber separator, the normal electrolyte dosage in one coin cell is 100 μL.
.....”

Referee comment #7:

The author should also investigate the interaction of long-chain polysulfide with the host material using polysulfide adsorption and symmetric cell studies (refer to this: Carbon 212 (2023) 118173, ACS Appl. Energy Mater. 2023, 6, 5, 3042–3051, Electrochimica Acta 422 (2022) 140531).

Author response #7:

We thank the referee for raising this important detail. We fully agree that the adsorption and conversion of long-chain polysulfides by host materials are of vital importance in KSBs and need to be further investigated. Following this suggestion, we provide a comprehensive experiment and discussion about the adsorption and conversion ability of host materials to long-chain polysulfides. The inspiring studies mentioned by the referee were cited in the revised manuscript.

First, to reveal the adsorption capabilities towards KPSs, we immersed the host materials of W_{SA}-W₂C@NC, W₂C@NC and NC into long-chain K₂S₆ solutions. The UV-vis spectra of three solutions after 1 h and 12 h were shown in Supplementary Fig. 17. Clearly, the W_{SA}-W₂C@NC host demonstrates the strongest adsorption capability towards K₂S₆, as proved by the complete disappearance of the adsorbance peak at 290 nm ascribed to K₂S₆. NC shows the weakest K₂S₆ adsorption, suggesting that it is the tungsten species providing K₂S₆ adsorption capability.

New Supplementary Figure 17. UV-vis spectra and the optical photos (inset) of different K_2S_6 solutions with different host materials immersed after **a** 1 h and **b** 12 h.

Second, the conversion performances of long-chain KPSs on different hosts were investigated by symmetric cell studies. The symmetric cells utilized $W_{SA}-W_2C@NC$, $W_2C@NC$, NC as electrodes, and 3 M KTFSI in TEGDME with 0.052 M K_2S_6 as electrolyte. We performed cyclic voltammetry analysis of the K_2S_6 symmetric cell at a scan rate of 25 mV s^{-1} in the potential range of -1.0 to 1.0 V to evaluate the catalytic ability of the host material towards long-chain KPSs conversion. The obtained CV curves (Supplementary Fig. 29) showed a pair of reversible redox peaks. The $W_{SA}-W_2C@NC$ electrode exhibited the highest redox current and the lowest polarization, suggesting best capability on accelerating the kinetics of long-chain KPSs conversion.

New Supplementary Figure 29. CV curves for at 25 mV s^{-1} K_2S_6 symmetric cells with NC, $W_2C@NC$, and $W_{SA}-W_2C@NC$ electrodes.

In addition, we performed a multi-scan rate CV measurement in the range of 10–100 mV s^{-1} for the symmetric cells. The corresponding CV curves are displayed in Figure R22. It is evident that with increasing scan rates, $\text{W}_{\text{SA}}\text{-W}_2\text{C@NC}$ electrodes consistently maintained a fast and sustained conversion of K_2S_6 with lowest polarization and highest response current.

Figure R22. Multi-rate scan CV curves for K_2S_6 symmetric cells with **a** NC, **b** $\text{W}_2\text{C@NC}$, and **c** $\text{W}_{\text{SA}}\text{-W}_2\text{C@NC}$ electrodes.

In order to accurately evaluate the reaction kinetics of long-chain KPSs affected by each host material, the key index b -values were calculated based on the multi-rate CV scan curves (as shown in Figure R23). $\text{W}_{\text{SA}}\text{-W}_2\text{C@NC}$ showed the largest b -value of 0.602, whereas of $\text{W}_2\text{C@NC}$ and NC demonstrate similar b -values (0.574 vs. 0.580). Therefore, it can be concluded that the synergy of W_{SA} and W_2C plays a key role for facilitating the redox process of long-chain KPSs.

Figure R23. Linear relationship of \log (current) vs. \log (scan rate) of redox peaks in the CV curves for the three electrodes.

The corresponding revisions are as follows in blue font.

“Electrochemical performances and sulfur redox mechanisms. $W_{SA}-W_2C@NC/S$ utilizing the host containing functional species of W_{SA} and W_2C is expected to be high-perform cathodes for KSBs. First, $W_{SA}-W_2C@NC$ and $W_2C@NC$ demonstrated much stronger adsorption capability towards KPSs than that of tungsten-free NC, as proved by the ultraviolet-visible (UV-vis) spectra of K_2S_6 solutions with different host materials (Supplementary Fig. 17).^{45, 46} This phenomenon agrees well with the calculation demonstrated in Fig. 1.

.....”

“.....

$W_{SA}-W_2C@NC/S$ cathode delivered *ca.* 25% higher response current in R_1 , which may probably be attributed to the facilitated long-chain KPSs migration across the carbon matrix by W_{SA} moieties. The most facile long-chain KPSs reduction kinetics enabled by $W_{SA}-W_2C$ can also be demonstrated by the CV profiles of K_2S_6 based symmetric cell (Supplementary Fig. 29).⁵⁷⁻⁵⁹

.....”

“.....

$W_{SA}-W_2C@NC/S$ delivered the highest response currents for $O_{1,2}$ peaks, highlighting the most favorable oxidation process from K_2S to long-chain KPSs. The K_2S_6 based symmetric cell tests further proved the most facile long-chain oxidation for $W_{SA}-W_2C@NC$ (Supplementary Fig. 29).”

Supporting Information

“.....

Adsorption measurements

The K_2S_6 solution for adsorption measurements was prepared by mixing potassium sulfide (K_2S) and sulfur with a molar ratio of 1:5 in dimethyl ether (DME). 5 mg $W_{SA}-W_2C@NC$, $W_2C@NC$, and NC were added into the 2 ml 0.02 M K_2S_6 solution, respectively, with the blank K_2S_6 solution as a reference.

Materials characterization

.....The ultraviolet-visible (UV-vis) spectra were performed by Shimadzu UV-2700 spectrophotometer.”

“.....

Symmetric cell measurements

The K_2S_6 electrolyte was fabricated by adding K_2S and sulfur (molar ratio corresponds to the nominal stoichiometry of K_2S_6) into the tetra ethylene glycol dimethyl ether (TEGDME) with 3 M potassiumbis(trifluoromethylsulfonyl)imide (KTFSI), and then stirring at 60 °C for 24 h. 100 μ L obtained K_2S_6 -contained electrolyte (0.052 M) with the identical anodes and cathodes of W_{SA} - $W_2C@NC$, $W_2C@NC$, and NC were assembled into the symmetric cells for the polysulfides conversion mechanism study.

.....”

References

45. *Carbon* **212**, 118173 (2023).
46. *Energy Stor. Mater.* **18**, 470 (2019).
57. *ACS Appl. Energy Mater.* **6**, 3042 (2023).
58. *Electrochim. Acta* **422**, 140531 (2022).
59. *Adv. Energy Mater.* **11**, 2100989 (2021).

REVIEWER COMMENTS

Reviewer #1 (Remarks to the Author):

The authors have addressed the reviewers comments satisfactorily and the manuscript has improved greatly in terms of quality. Hence I recommend the work to be published.

Reviewer #2 (Remarks to the Author):

Authors have professionally revised the manuscripts according to my comments, and their revisions with the additional experiments and calculations also have significantly improved this interesting work. Therefore, I strongly recommend this work can be accepted as it is.

Reviewer #3 (Remarks to the Author):

The work reported by W. Song et al. used WSA-W2C-NC composite as a sulfur for improving the performance of KSBs. The manuscript has been significantly improved after including the suggestions from the referees making it more impactful. Therefore, I recommend publishing this article after addressing the formatting and a few technical comments as provided below.

Comments for the author:

1. In abstract line 22: instead of high-perform I feel high-performance or high-performing will be more appropriate. Similarly, in line 232: the product remained should be product retained. For smooth reading by the reader, tables in the supporting information should also be arranged in order as they are cited in the manuscript, not after the images.
2. As the authors performed N₂-sorption isotherm, it would be better for the reader, if the author could include the surface area report to confirm the effective sulfur infusion in the samples.
3. In Figure S17, the author should comment on the reason for the same extent of absorption for K2S6 on WSA-W2C@NC and W2C@NC surface after 12 hours.
4. The author should also explain the three peaks observed during anodic and cathodic in the CV curve shown in Figure 3a.

Response Letter for Manuscript “Synergy of facilitated migration and catalytic conversion towards potassium polysulfides enables high performance potassium-sulfur batteries (NCOMMS-23-28977A)”

We are grateful for the referee’s constructive reviews. The manuscript has been revised in accordance with the comments. The point-to-point responses are detailed in this Response Letter. The corresponding changes and revisions are highlighted in **blue font** in the Response Letter, the revised manuscript, figures (legends) and Supporting Information.

Reviewer #3:

General comments:

The work reported by W. Song et al. used $W_{SA}-W_2C-NC$ composite as a sulfur for improving the performance of KSBs. The manuscript has been significantly improved after including the suggestions from the referees making it more impactful. Therefore, I recommend publishing this article after addressing the formatting and a few technical comments as provided below.

Author response:

We sincerely appreciate for the reviewer’s carefully reading of our work and the highly constructive comments. We have addressed all the formatting and technical comments in this version of Response Letter. The revised contents are all in blue font.

Referee comment #1:

In abstract line 22: instead of high-perform I feel high-performance or high-performing will be more appropriate. Similarly, in line 232: the product remained should be product retained. For smooth reading by the reader, tables in the supporting information should also be arranged in order as they are cited in the manuscript, not after the images.

Author response #1:

We sincerely thank the referee for these comments, and we fully agree with these valuable suggestions. Accordingly, we have revised the phrasings aforementioned.

Also, we have rearranged the orders of Figures and Tables in the Supporting Information according to their cited position in the main manuscript.

The revisions are in blue font as follows.

“

Abstract

.....Here, we report a strategy of synergizing the facilitated migration and catalytic conversion towards potassium polysulfides (KPSs) for building **high-performance** KSBs.....

”

“.....

W_{SA} - W_2C @NC/S utilizing the host containing functional species of W_{SA} and W_2C is expected to be **high-performance** cathodes for KSBs.

.....”

“.....

Employing the aforesaid hosts, W_{SA} - W_2C @NC/S, W_2C @NC/S, and NC/S were prepared by melting-impregnation method. The **products retained** the morphologies of the pristine hosts (Supplementary Fig. 9).

.....”

The order of Figures and Tables in the revised Supporting Information is as below.

Supplementary Figure 1.

Supplementary Table 1.

Supplementary Figure 2.

Supplementary Figure 3.

Supplementary Figure 4.

Supplementary Figure 5.

Supplementary Figure 6.

Supplementary Figure 7.

Supplementary Figure 8.

Supplementary Figure 9.

Supplementary Figure 10.

Supplementary Figure 11.

Supplementary Figure 12.

Supplementary Figure 13.

Supplementary Figure 14.

Supplementary Figure 15.

Supplementary Figure 16.

Supplementary Figure 17.

Supplementary Figure 18.

Supplementary Figure 19.

Supplementary Table 2.
Supplementary Figure 20.
Supplementary Table 3.
Supplementary Figure 21.
Supplementary Figure 22.
Supplementary Table 4.
Supplementary Figure 23.
Supplementary Figure 24.
Supplementary Figure 25.
Supplementary Figure 26.
Supplementary Table 5.
Supplementary Figure 27.
Supplementary Figure 28.
Supplementary Figure 29.
Supplementary Figure 30.
Supplementary Figure 31.
Supplementary Figure 32.

Referee comment #2:

As the authors performed N₂-sorption isotherm, it would be better for the reader, if the author could include the surface area report to confirm the effective sulfur infusion in the samples.

Author response #2:

We thank the referee for this excellent suggestion. We fully agree that the changes in surface areas can confirm the effective sulfur infusion in the hosts. To address this comment, we supplemented discussions about the surface areas and their correlation with sulfur infusion in the revised manuscript. Basically, the surface areas of the hosts significantly decreased after the sulfur impregnation, which strongly confirm the accommodation of sulfur in the porosity of hosts.

The corresponding revisions are as below.

“.....

According to the nitrogen adsorption-desorption isotherms analyses (Supplementary Fig. 16), the significantly decreased surface areas of the composites as compared to pristine hosts (e.g. 15.96 m² g⁻¹ of W_{SA}-W₂C@NC/S versus 706.60 m² g⁻¹ of W_{SA}-W₂C@NC) confirmed the effective sulfur infusion. Moreover, the sulfur hosts all provide microporosities for sulfur accommodation, which should be the essential

reason for the amorphous structure of the impregnated sulfur.

.....”

Supplementary Figure 16. **a** Nitrogen adsorption-desorption isotherms and **b** pore size distribution of NC, W₂C@NC and W_{SA}-W₂C@NC. **c** Nitrogen adsorption-desorption isotherms and **d** pore size distribution of NC/S, W₂C@NC/S and W_{SA}-W₂C@NC/S. BET surface values are annotated after the sample IDs.

Referee comment #3:

In Figure S17, the author should comment on the reason for the same extent of absorption for K₂S₆ on W_{SA}-W₂C@NC and W₂C@NC surface after 12 hours.

Author response #3:

We thank the referee for raising this good point about the same extent of K₂S₆ absorption by W_{SA}-W₂C@NC and W₂C@NC after 12 hours. As indicated by the zero intensities of K₂S₆ peaks (290 nm) in the spectra (overlap with the spectrum of pure DME solvent), all K₂S₆ species were captured by W_{SA}-W₂C@NC and W₂C@NC. The reason of this phenomenon is that it is the W₂C nanocrystals providing the active sites for strong potassium polysulfide adsorption. Both W_{SA}-W₂C@NC and W₂C@NC possess dense W₂C nanocrystals, which can thoroughly adsorb the K₂S₆ species in the

solution after extended time of 12 hours.

Supplementary Figure 17. UV-vis spectra and the optical photos (inset) of different K_2S_6 solutions with different host materials immersed after **a** 1 h and **b** 12 h. Spectrum of pure DME solvent is also included for comparison.

The revision about this point is as below.

“.....

First, $W_{SA}-W_2C@NC$ and $W_2C@NC$ demonstrated much stronger adsorption capability towards KPSs than that of tungsten-free NC, as proved by the ultraviolet-visible (UV-vis) spectra of K_2S_6 solutions with different host materials (Supplementary Fig. 17).^{45, 46} This phenomenon agrees well with the calculation demonstrated in Fig. 1. After 12 hours, all K_2S_6 species in the solution were thoroughly adsorbed by $W_{SA}-W_2C@NC$ and $W_2C@NC$ hosts, as evidenced by the corresponding spectra overlapping with pure DME solvent.

.....”

Referee comment #4:

The author should also explain the three peaks observed during anodic and cathodic in the CV curve shown in Figure 3a.

Author response #4:

We thank the referee for this valuable suggestion. The cathodic and anodic peaks in the CV curves in Figure 3a are highly important for analyzing the kinetics of different steps of potassium polysulfides K_2S_6 conversion. We resolved these cathodic/anodic peaks

by *in-situ* XRD (Figure 3f), *ex-situ* SAED (Figure 3g) and *ex-situ* XPS (Figure 3h) characterizations. Basically, the R₁/R₂ peaks in the cathodic scanning process were ascribed to the successive conversion of long-chained KPSs to short-chained KPSs, and R₃ was related to the reduction of short-chained KPSs to end member of K₂S. The O₃/O₂ and O₁ peaks in the anodic scan correspond to the inverse oxidation reactions. Specifically, O₁ and O₃/O₂ should be ascribed to the oxidation of solid-state K₂S to short-chain KPSs and the further oxidation of long-chain KPSs, respectively.

The relevant discussions related to this point in the manuscript are as below in blue font.

“.....

Reduction peaks at 1.594 (R₁), 1.329 (R₂), and 0.761V (R₃) appeared in the 2nd cycle (Fig. 3a). In the anodic scan, oxidation peaks present at 1.844 (O₁), 2.159 (O₂), and 2.272V (O₃). W₂C@NC/S and NC/S exhibit similar shapes of CV curves as W_{SA}-W₂C@NC/S, but deliver lower response currents and larger polarizations. This phenomenon suggested the faster sulfur redox kinetics for W_{SA}-W₂C@NC/S than the other two cathodes. According to the systematic spectroscopy data, the sulfur redox process can be resolved. With regard to the CV curve in Fig. 3a, R₁/R₂ should be ascribed to the successive conversion of long-chain to short-chain KPSs, and R₃ is associated with the reduction of short-chain KPSs into K₂S. In the anodic scan, O₃/O₂, and O₁ should be ascribed to the long-chain KPSs oxidation and solid-state KPSs oxidation, respectively. Comparing the peak intensities in the CV curves, the prominent long-chain KPSs conversion for NC/S, and facile short-chain KPSs conversion for W₂C@NC/S can be well distinguished. Meanwhile, W_{SA}-W₂C@NC/S cathode exhibited the most facile kinetics in all the steps of long-/short-chain KPSs conversion.

.....”